



# Temporal and spatial variability in surface roughness and accumulation rate around 88° S from repeat airborne geophysical surveys

Michael Studinger[1], Brooke C. Medley[1], Kelly M. Brunt[2,1], Kimberly A. Casey[3,1], Nathan T. Kurtz[1], Serdar S. Manizade[4,5], Thomas A. Neumann[1], and Thomas B. Overly[2,1]

[1]NASA Goddard Space Flight Center, Greenbelt, MD, USA
[2]Earth System Science Interdisciplinary Center (ESSIC), University of Maryland, College Park, MD, USA
[3]U.S. Geological Survey, Reston, VA, US
[4]Amentum Services Inc., Wallops Island, VA, USA
[5]NASA Wallops Flight Facility, Wallops Island, VA, USA

*Correspondence to*: Michael Studinger (michael.studinger@nasa.gov)

**Abstract.** We use repeat high-resolution airborne geophysical data consisting of laser altimetry, snow and Ku-band radar and optical imagery acquired in 2014, 2016 and 2017 to analyze the spatial and temporal variability in surface roughness, slope, wind deposition, and snow accumulation at 88° S as this is a bias validation site for ICESat-2 and may be a potential validation site for CryoSat-2. We find significant small–scale variability (< 10 km) in snow accumulation based on the snow radar subsurface stratigraphy, indicating areas of strong wind redistribution are prevalent at 88° S. In general, highs in snow accumulation rate correspond with topographic lows resulting in a negative correlation coefficient of $r^2 = -0.32$ between accumulation rate and MSWD (Mean Slope in the mean Wind Direction). This relationship is strongest in areas where the dominant wind direction is parallel to the survey profile, which is expected as the geophysical surveys only capture a two-dimensional cross section of snow redistribution. Variability in snow accumulation appears to correlate with variability in MSWD. The correlation coefficient between the standard deviations of accumulation rate and MSWD is $r^2 = 0.48$ indicating a stronger link between the standard deviations than the actual parameters. Our analysis shows that there is no simple relationship between surface slope, wind direction and snow accumulation rates for the entire survey area. We find high variability in surface roughness derived from laser altimetry measurements on length–scales smaller than 10 km, sometimes with very distinct and sharp transitions. Some areas also show significant temporal variability over the course of the 3 survey years. Ultimately, there is no statistically significant slope–independent relationship between surface roughness and accumulation rates within our survey area. The observed correspondence between the small–scale temporal and spatial variability in surface roughness and backscatter, as evidenced by Ku-band radar signal strength retrievals, will make it difficult to develop elevation bias corrections for radar altimeter retrieval algorithms.

## 1 Introduction

Polar ice sheets play a critical role in Earth's climate system. Measurements from satellites and aircraft reveal that the ice sheets of Greenland and Antarctica are changing at an accelerating rate suggesting increasing rates of global sea–level rise as the ice sheets melt (e.g., Vaughan et al., 2013). To project future rates of sea–level rise, numerical models of an ice sheet's response to climate forcing require input data of the surface mass balance and its spatial and temporal variability. Observing changes in ice–surface elevation from satellite and airborne platforms has long been recognized as a powerful tool for assessing and quantifying ice sheet mass balance (e.g., Abdalati et al., 2010; Krabill et al., 2000; Thomas and Investigators, 2001; Zwally et al., 2002). The southern convergence of all Ice, Cloud, and land Elevation Satellite-2 (ICESat-2, (Markus et al., 2017)) and CryoSat-2 (Wingham et al., 2006) ground reference tracks at 88° S is in a region of low snow accumulation (7–10 cm annual water equivalent)





(McConnell et al., 1997; Mosley-Thompson et al., 1999; Winski et al., 2019) and low surface slope (0.11° ± 0.10°, Fig. A1a) (Helm et al., 2014). Because of the density of tracks, the small impact of surface processes, and the region's relative quiescence,

88° S is the primary ICESat-2 land–ice validation site in the southern hemisphere (Brunt et al., 2019a; Brunt et al., 2019b). Both radar and laser altimeters are potentially prone to elevation biases related to surface roughness and slope. For laser altimeters such as ICESat-2, increased surface roughness causes broadening of the return signal, which can cause elevation biases up to 0.2 m (Smith et al., 2019). When surface roughness changes seasonally the elevation biases will also change with time (Smith et al., 2019). For radar altimeters such as CryoSat-2, smoother surfaces will have larger return signal strength compared to rougher

surfaces which also changes the shape of the return waveform potentially causing elevation biases (Kurtz et al., 2014). Since radar altimeters penetrate below the ice surface volume backscatter from subsurface firn will also impact the return signal waveform (e.g., Nilsson et al., 2016). Radar extinction with depth depends on the dielectric permittivity of firn, which is primarily a function of firn density (Kovacs et al., 1995). Changes in firn density are often related to changes in snow accumulation rates (e.g., Grima et al., 2014) making radar elevation biases potentially a function of spatial changes in accumulation rates. Furthermore, wind-

induced anisotropic features of firn can introduce azimuth depending elevation biases (Armitage et al., 2014).

Previous Antarctic studies have reported relationships between surface slope, roughness and snow accumulation rates (e.g., Arcone et al., 2012; Dattler et al., 2019; Fahnestock et al., 2000; Grima et al., 2014; Hamilton, 2004; King et al., 2004). To better understand potential correlations between altimetry elevation biases and geophysical parameters of the ice surface, we are specifically studying the spatial and temporal variability of surface roughness and accumulation rate over the ICESat-2 validation site at 88° S. We use

repeat high–resolution airborne data consisting of laser altimetry, snow and Ku–band radar and natural color imagery acquired as part of the National Aeronautics and Space Administration's (NASA) Operation IceBridge (OIB) mission to analyze spatial and temporal variability in surface roughness, slope, accumulation rate, and Ku-band radar backscatter along a 1400 km circle around 88° S (Fig. 1).

**Figure 1: Location map with survey area around 88° S (yellow line). Surface elevation is from Helm et al., (2014). Rock outcrops are marked in red. SP marks the geographic South Pole, TD is Titan Dome and FIS is Filchner Ice Shelf.**





## 2 Data sets

We use airborne geophysical data collected during 6 NASA OIB survey flights in 2014, 2016 and 2017. The data consists of high-resolution laser altimetry, natural color imagery and snow and Ku-band radar data and is available from the National Snow and Ice

Data Center (NSIDC).

### 2.1 Laser altimeters

#### 2.1.1 Airborne Topographic Mapper (ATM)

The ATM is a conically-scanning laser altimeter that measures the surface topography of a swath beneath the aircraft at a 15° off–nadir angle (Krabill et al., 2002). The range from the laser altimeter to the surface is converted to geographic position by integration

with platform Global Positioning System (GPS) and attitude/Inertial Measurement Unit (IMU) measurement subsystems. The conical scan geometry results in a near–constant angle of incidence and intersecting laser footprints allow to determine pointing biases over any type of surface (Harpold et al., 2016; Martin et al., 2012). The two generations of instruments used in this study, T4 and T6 have a pulse repetition frequency of 3,000 Hz, a wavelength of 532 nm, and a pulse width of 6 ns full width at half maximum (FWHM). The ATM scanner has a swath width of 240 m at a nominal flight elevation of 460 m above ground level

(AGL) and a footprint diameter of ~0.8 m. As a result of the conical scan pattern, the density of spot elevation measurements varies across the swath from 0.03 footprints m$^{-2}$ at the center to 0.37 footprints m$^{-2}$ at the edge. At a nominal aircraft speed of 130 m s$^{-1}$ the average spacing between point elevation measurements is ~5 m in the center of the scan and <1 m near the edge. The vertical accuracy of an individual laser spot measurement is estimated to be 7 cm with a vertical shot-to-shot precision of 3 cm (Martin et al., 2012). We use both the L1B and L2 (ICESSN) data products (Studinger, 2013, updated 2018, 2014, updated 2018).

#### 2.1.2 Riegl LMS–Q240i laser altimeter

The University of Alaska, Fairbanks (UAF) operates a commercially available Riegl LMS–Q240i airborne laser scanner together with IMU and dual–frequency GPS subsystems for attitude and precise position. The system is a near–infrared linear, unidirectional scanner that scans the surface in parallel lines. The system acquires measurements at 10,000 Hz with a footprint size of 1.0 m at 460 m AGL and at a ±30° off–nadir scan angle. The average spacing of laser footprints both, along track and across is ~1 m at 460

m AGL and an ground speed of 85 m/s (Johnson et al., 2013; Larsen, 2010, updated 2018). Results over 20% of our study area at 88° S show that 2 flights from a 2017 UAF laser altimetry survey had a <10 cm bias and a surface measurement precision of <10 cm (Brunt et al., 2019a).

### 2.2 Snow and Ku-band radar

The snow radar is an ultra–wideband microwave radar that operates over the 2–8 GHz frequency range and is developed and

operated by the Center for Remote Sensing of Ice Sheets (CReSIS) at the University of Kansas (KU). The system is a frequency–modulated continuous–wave (FMCW) radar that images the stratigraphy in the upper ~40 m of the ice sheet with a bandwidth-limited range resolution of 2.6 cm in firn with a density of 550 kg/m$^3$ and 3.8 cm in air (Leuschen, 2014, updated 2018; Panzer et al., 2013; Rodriguez-Morales et al., 2014). This is below the resolution necessary to resolve annual layers in regions with very low annual snow accumulation such as our survey area. At the nominal flight elevation of 460 m AGL the snow radar has a footprint

size of approximately 10 m across track and 14.5 m along track (Panzer et al., 2013). A 9–by–4 (range bin–by–trace) median filter is applied to the data to minimize noise. The Ku-band radar altimeter is identical in design but operates over the frequency range of 12–18 GHz for mapping subsurface stratigraphy in the upper 10 m of polar firn (Gomez-Garcia et al., 2012; Paden et al., 2014,



updated 2018; Rodriguez-Morales et al., 2014). Since both radars have the same bandwidth (6 GHz) the bandwidth-limited range resolution of the Ku-band radar is the same as the snow radar.

**2.3 Digital Mapping System natural color imagery**

The Digital Mapping System (DMS) is a digital camera that acquires natural color, high–resolution images at 10 cm pixel size at the nominal flight elevation of 460 m AGL (Dominguez, 2010, updated 2018) (Fig. 2). The camera is operated by NASA's Airborne Sensor Facility located at the Ames Research Center. Images are approximately 380 m across swath and 570 m along swath and cover the entire ATM swath width. A combined IMU and GPS system for precise position and attitude information is part of the instrument package. DMS images are acquired with overlap between consecutive images to ensure data continuity. The difference in geolocation between distinct elongated snow surface features between overlapping, orthorectified images is on the order of several meters. These elongated topographic snow features are called sastrugi. The DMS images are referenced to the RADARSAT 200 m Digital Elevation Model (DEM) (Liu et al., 2015).

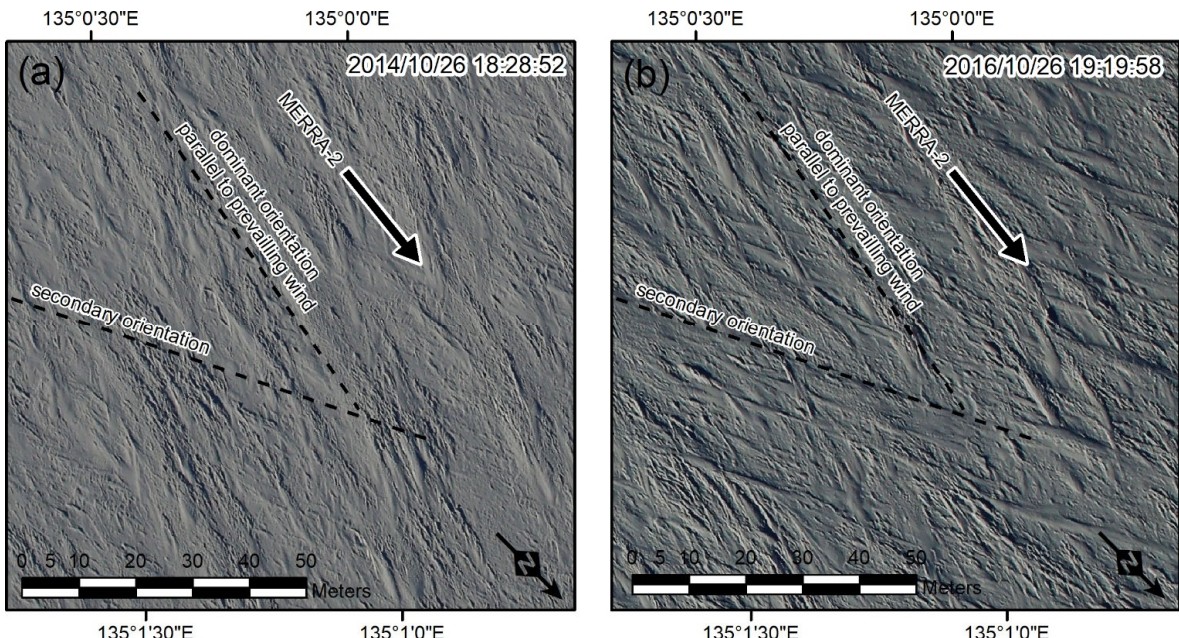

**Figure 2: DMS natural color images of the same area at 88° S and 135° E. The two aerial images are nadir–looking, geolocated and orthorectified, color photographs of a sun illuminated snow surface taken from 460 m AGL. The two photographs were taken on the same day of year two years apart. The low–angle sun illuminates the elongated, elevated surface features (sastrugi) facing the sun and creates dark shadows in the opposite direction behind the elevated features. For the 2014 image the sun is 11.5° above the horizon (accounting for refraction through the atmosphere) and for the 2016 image it is 12.1°. The orientations of sastrugi, indicated by dashed lines, is determined by visually following transitions of elongated bright, sun–illuminated features and corresponding shadows. Both images show a dominant sastrugi orientation parallel to the 26 year averaged 10 m wind field from MERRA-2 (e.g., Gelaro et al., 2017) and a secondary orientation that appears to be less pronounced in 2014 compared to 2016.**

**2.4 Survey flights**

Between 2014 and 2017 six NASA OIB airborne geophysical survey flights were completed to acquire data around 88° S (Table 1). Two flights are necessary to complete the entire small circle around 88° S with the platforms and bases of operations used. Snow and Ku–band radar data and DMS images are only available for 2014 and 2016 years. The combination of simultaneous laser altimetry, snow radar stratigraphy and natural color imagery on a regional scale provides a unique data set to study small scale deposition and erosional processes and their temporal and spatial variability.





| Date | Longitude Segment | Aircraft | Laser | Snow & Ku-band Radars | Camera |
|------|-------------------|----------|-------|-----------------------|--------|
| 2014/10/23 | 110° E to 70° W | DC–8 | ATM–T4 | KU CReSIS | DMS |
| 2014/10/26 | 70° W to 110° E | DC–8 | ATM–T4 | KU CReSIS | DMS |
| 2016/10/26 | 70° W to 110° E | DC–8 | ATM–T6 | KU CReSIS | DMS |
| 2016/11/15 | 110° E to 70° W | DC–8 | ATM–T6 | KU CReSIS | DMS |
| 2017/11/30 | 100° E to 10° W | DC–3 | UAF Riegl | n/a | n/a |
| 2017/12/03 | 30° W to 150° W | DC–3 | UAF Riegl | n/a | n/a |

**Table 1: Science instruments and airborne platforms of 6 NASA OIB aerogeophysical survey flights at 88° S.**

## 3 Ice surface roughness

### 3.1 Background

The surface roughness of polar ice sheets is primarily a result of ice dynamics and surface–atmosphere interactions on varying temporal and spatial scales. In general, ice flow over rugged bedrock topography causes roughness features that can extend from

several hundreds of kilometers to a few kilometers depending on ice thickness, flow speed and basal conditions (e.g., Smith et al., 2006, and references herein). These large–scale variations in ice surface topography caused by ice dynamics are not the topic of this analysis. Here, we focus on small–scale surface roughness or surface texture that spans from several meters to hundreds of meters and is primarily the result of ice–atmosphere interactions, such as wind deposition and wind–induced ablation or erosion, the predominant types of surface-atmosphere interactions in the area of 88° S. Elongated topographic snow features, called sastrugi,

are the dominant form of small–scale surface roughness in the interior of polar ice sheets. Sastrugi are known to form parallel to the prevailing wind direction and their orientation can therefore be used to infer time–averaged prevailing wind directions (e.g., Bromwich et al., 1990; Gow, 1965). Figure 2 shows natural color DMS images of the same area at 88° S and 135° E taken two years apart. The dominant sastrugi orientation matches the 26 year average (1980 – 2016) of the 10 m wind direction from MERRA-2 (Modern-Era Retrospective Analysis for Research and Applications, (e.g., Gelaro et al., 2017)) (Fig. 3a). The ice surface on the

East Antarctic plateau often has a dominant sastrugi orientation with sometimes two or three populations of sastrugi forming a crossing network of ridges that reflects seasonal changes in wind orientation (e.g., Warren et al., 1998) (Fig. 2). These seasonal changes are not captured in our averaged MERRA-2 wind direction. The good agreement between the dominant sastrugi orientation and the MERRA-2 long–term average, however, suggests that a single dominant wind direction is a good representation of the conditions in the survey area (Fig. 2).

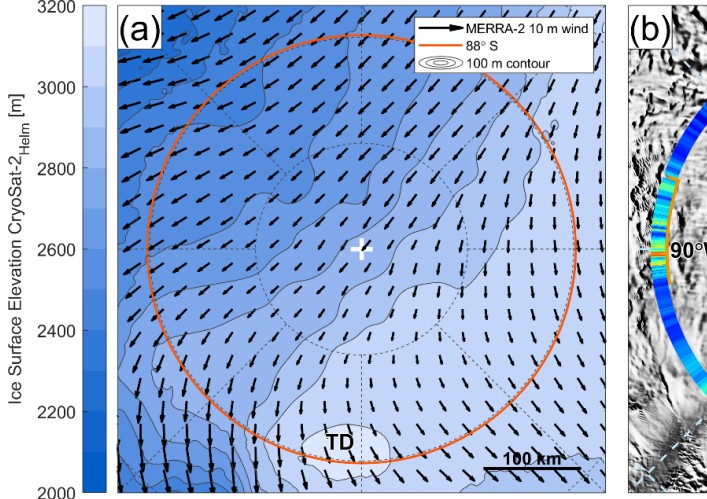

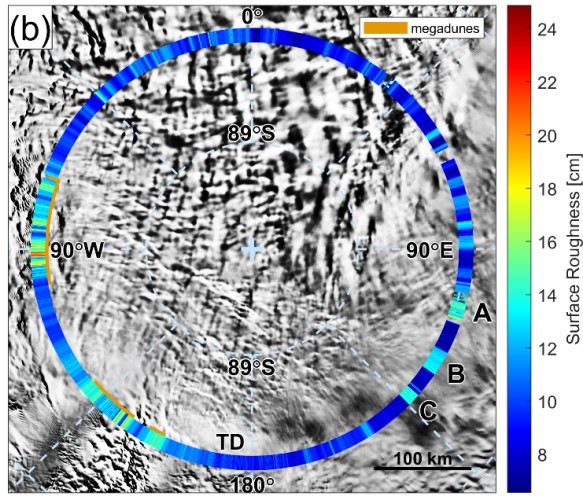




**Figure 3: a) Ice surface elevation from CryoSat-2 (Helm et al., 2014) and 26 year 10 m average wind speed from MERRA-2 (e.g., Gelaro et al., 2017). Red line marks location of 88S° S airborne geophysical data collection. TD is Titan Dome. b) Ice surface roughness from 2014 ATM data (Studinger, 2014, updated 2018). Background image is MODIS Mosaic of Antarctica 2013 – 2014 (Haran et al., 2014) using MODIS data between 11/2013 and 03/2014. Locations of megadune fields are marked in orange and roughness features A, B, and C refer to Fig. 4b.**

In recent decades, especially in the rapidly warming West Antarctic region, synoptic heat– and moisture–bearing storms have reached the South Pole area (e.g., Harris, 1992; Nicolas and Bromwich, 2011). Such storms and cyclonic events have been less common, though still occur in the interior of East Antarctica (e.g., Gorodetskaya et al., 2014; Hirasawa et al., 2000). Individual sastrugi can be eroded and reform during a single storm (Warren et al., 1998). Most changes, however, occur as a result of seasonal changes between summer and winter months (e.g., Gow, 1965). Gow (1965) shows that sastrugi form during winter months resulting in a rough surface, and subsequently get eroded during the summer by sublimation and deflation at South Pole. This effect mostly results in a flattening of the subsurface layer stratigraphy and therefore does not affect our surface roughness results.

## 3.2 Relevance of surface roughness and slope for altimetry and surface mass balance

The surface roughness and slope of ice sheets affect several processes that are relevant for ice sheet mass balance (e.g., van der Veen et al., 2009, and references herein) (Gow, 1965). King et al., (2004) describe small scale variations in accumulation rate on the order of 1 km that appear to be associated with wind–borne redistribution as a function of slope. Hamilton (2004) found significant variability in snow accumulation rates due to the interaction of prevailing winds with meter–scale surface topography; where, for example, a concave depression can receive up to 18% more accumulation than adjacent steeper snow surface topography. Similarly, Arcone et al. (2012) mapped accumulation patterns in East Antarctica that are created by wind-blown deposition on windward and leeward slopes. Slope–dependent accumulation are also related to spatial variations in firn density (Grima et al., 2014) which impacts mass balance estimates from altimetry data. Small–scale roughness contributes to noise in firn core records and therefore accumulation rate estimates (van der Veen et al., 1998; van der Veen et al., 2009). Studies by van der Veen et al. (1998; 2009) used ATM roughness estimates over Greenland to determine the uncertainty in water equivalent (w.e.q.) accumulation estimates from shallow firn cores.

Surface roughness also affects the albedo and bidirectional reflectance distribution function (BRDF) of ice sheets (Leroux and Fily, 1998; Warren et al., 1998). Nolin et al. (2002) used ATM roughness estimates to calibrate Multi–angle Imaging SpectroRadiometer (MISR) roughness estimates, and Nolin and Payne (2007) derived then relationships between ice surface roughness and near–infrared albedo using ATM and MISR data. Ongoing satellite and modelling investigations on radiative impacts of surface roughness and sastrugi continue to illuminate angular relationships and parameterizations that can be key to quantifying BRDF and albedo sensitivities in ice surface studies (e.g., Corbett and Su, 2015; Kokhanovsky and Zege, 2004; Larue et al., 2019). As ice sheet surface roughness mapping and modeling capabilities improve, it will be possible to more accurately include the radiative effects of surface roughness. Surface roughness furthermore affects thermodynamic fluxes because it affects boundary layer processes through the aerodynamic roughness and therefore the surface energy balance (Boisvert et al., 2017; Chambers et al., 2019; Nolin and Mar, 2018; Palm et al., 2017).

## 3.3 Surface roughness estimates

There are several diverse approaches to quantifying topographic irregularity or surface roughness (e.g., Smith, 2014, and references herein). In general, roughness metrics are not only scale and orientation–dependent, but also impacted by the spatial resolution, footprint size and sample spacing of the input data. One commonly used metric for surface roughness is the standard deviation σ of small–scale elevation fluctuations from a mean or de–trended surface in a given area or over a given length (e.g., Das et al., 2013; Smith, 2014, and references herein). In order to minimize potential effects from anisotropy in surface roughness over





different length scales and orientations we have calculated surface roughness over an area roughly square in size common to both laser altimeters. Individual spot elevation measurements are binned into 0.06° longitude segments (240 m in length at 88°S). We then fit a 3$^{rd}$ order polygon regression model through all spot elevation measurements within a longitude segment. We define the standard deviation σ of the residuals as a metric for surface roughness. Because of the very shallow surface slopes around 88° S

and the small area of the selected longitude bins there is no significant difference between removing a bilinear trend, quadratic surface or a 3$^{rd}$ order polygon.

For estimation of surface roughness from snow radar data, we pick an initial surface for each trace (every ~5 meters along track) by finding the maximum slope in the radar return power across 9 range bins. A radar trace records the reflected returns from a transmitted radar signal in discrete time intervals called range bins. The size of a range bin in firn with a density of 550 kg/m$^3$ is

1.8 cm. Starting at the initial surface pick, we keep sliding the surface pick one range bin deeper (or later in time) while the slope for the range bin remains above 3 standard deviations of the mean, which provides our final surface. Next, surface picks that lie outside of a 15–range bin window from a smoothed surface are discarded and set to the smoothed surface, however, very few data points are discarded. Surface roughness is estimated from residuals to the smoothed surface fit. Specifically, surface roughness for a given location is calculated as the standard deviation in surface range–bin residuals for locations within a 250–meter radius. This

radius was selected to ensure consistency with laser-altimeter–derived roughness values. Finally, the range–bin roughness is converted to heights by using the radar wave velocity in air.

For a closer look at very fine–scale spatial and temporal changes in surface roughness around 88° S we use the ATM Level 2 ICESSN roughness estimates. The ATM Level 2 ICESSN data product includes slope and roughness estimates in overlapping 80 m × 80 m platelets across the swath (Studinger, 2014, updated 2018). The root mean square of the residuals of a plane fit through

the platelets is an estimate of the surface roughness. Removing the mean results in the root mean square being equivalent to the standard deviation σ.

Figure 4 shows the surface roughness estimates around 88° S latitude from three different instruments and over the course of 3 different years. In general, there is good agreement between the roughness estimated using the ATM laser altimeter and the synchronous roughness estimates from the snow radar (Fig. 4 b, c). Because of the smaller footprint size and higher sampling

density the ATM laser–derived roughness estimates are slightly larger than the radar estimates, with few exceptions. The radar estimates also reveal more scatter, probably caused by the much lower range resolution of the radar compared to the ATM laser altimeter. The mean difference between the laser minus snow radar roughness estimates is 0.6 ± 1.9 cm for 2014 and 1.6 ± 2.1 cm for 2016; however, the spatial patterns, which are of main interest for this study, are nearly indistinguishable. There are no obvious spatial patterns in the roughness difference between laser and radar that would reflect a geophysical signal. Because of the higher

point density roughness derived from the UAF Riegl system is larger than the ATM roughness estimates. The mean difference between the 2017 UAF minus 2016 ATM laser estimates is 1.2 ± 2.4 cm.

A 370 km long segment between 150°W and 100°E was repeated in 2017 within 3 days with the same instrument. It is unlikely that the surface was significantly altered within 3 days and therefore the difference between the two estimates can be used as an approximate estimate of the instrument–specific accuracy and precision of the laser–derived roughness estimates. The mean surface

roughness for the first east-bound flight is 9.5 ± 1.5 cm and 10.1 ± 2.0 cm for the later west-bound flight. The mean μ of the difference in surface roughness between the two 2017 flights is 0.02 cm with a σ of 0.7 cm. For comparison, a separate study by Das and others (2013) over Dome Argus with a Riegl LMS–Q240i scanner show a similar range of roughness as our measurements around 88° S.



**Figure 4: a) Ice surface elevation from 2014 ATM data (Studinger, 2014, updated 2018) and bedrock topography from 2014 CReSIS Multichannel Coherent Radar Depth Sounder (MCoRDS) data (Leuschen et al., 2010, updated 2018). b) Surface roughness from 2014 ATM laser and snow radar data. A 14 minute disk failure of the snow radar in 2014 resulted in a data gap over the megadunes. c) Same for 2016. d) Surface roughness from 2017 UAF laser data e) Surface slope from 2014 and 2016 ATM ICESSN data using only the nadir platelet (Studinger, 2014, updated 2018). Heavy dashed lines in b) – e) mark megadune areas and thin dashed lines in b) indicate distinct abrupt changes in surface roughness in 2014.**

### 3.4 Surface roughness, slope and elevation around 88° S

In order to distinguish ice surface roughness features caused by ice dynamics from roughness features that are a result of ice–atmosphere interactions, the proximity of the ice surface to bedrock topography and bedrock roughness must be understood (Fig. 4 a). The large ice thickness (Fig. 4) and slow ice flow velocities (< 10 m yr⁻¹ (Mouginot et al.)) in the survey area, combined with the small window size we use to calculate the roughness make it unlikely that any of the roughness features we observe are ice dynamic related. Thus, we interpret the roughness characteristics shown in Figs. 3 and 4 as caused by ice–atmosphere interactions.



Figure 3b and 4 show several spatially coherent segments with distinct roughness characteristics around 88° S that appear not to be related to ice dynamics. In general, the surface roughness estimated from snow radar and laser altimetry data varies between 2 and 25 cm. The smoothest surface in 2016 and 2017 is between 175° W and 60° E and includes Titan Dome. The smooth segment

also coincides with the highest ice surface elevations and shallowest surface slopes around 88° S (Fig. 4).

The segment between 70° W and 100° W shows a pronounced increase in roughness in 2014 (Fig. 4b). The MODIS Mosaic of Antarctica (MOA, (Haran et al., 2014)) shows that this segment is near the edge of a megadune field that is mostly north of 88° S (Fig. 3b). Megadunes are long–wavelength surface ripples (Fahnestock et al., 2000) with amplitudes on the order of a few meters (peak to trough) and wavelengths of several kilometers (Fahnestock et al., 2000; Scambos and Fahnestock, 1998). The typical

elevation pattern of megadunes is not visible in the 88° S laser altimetry data. The likely reasons for this are that the airborne geophysical data was collected at the edge of the dune field and the orientation of the dune crests is subparallel to the airborne geophysical data. The orientation of dune crests between 70° W and 100° W is approximately perpendicular to the prevailing surface wind direction from MERRA-2 (Fig. 3a) consistent with findings from Fahnestock et al. (2000).

A second megadune field can be seen between 130° W – 145° W and 150° W – 155° W (Fig. 3 and Fig. 4). In 2017 this dune field

appears to have less of roughness anomaly compared to the 2014 and 2016 data. Data for MOA was collected between 11/2013 and 03/2014. The temporal stability of megadune fields remains poorly understood. Fahnestock et al. (2000) found 60 m of dune migration over a 34 year period. Since our survey area is near the edge of the dune field we cannot rule out that over the course of 4 year the edge of the dune field has migrated out of the coverage of the airborne geophysical data.

The slope of the ATM ICESSN nadir platelets shows many distinct peaks that are aligned very well between the 2014 and 2016

data, indicating that these features are stable in location (Fig. 4e). The mean μ and standard deviation σ of the surface slope around 88° S is 0.20° ± 0.16° and never exceeds 1.5°. Together with the low snow accumulation rate and low ice surface velocities this makes it an ideal area for calibration and validation of spaceborne altimeters (Brunt et al., 2019a; Brunt et al., 2019b).

### 3.5 Temporal changes in surface roughness

We use ATM Level 2 ICESSN roughness estimates for a closer look at multiyear variability in surface roughness characteristics

around 88° S (Fig. 5) because they are calculated over 80 m × 80 m platelets. During 2014, several spatially coherent segments show an approximately two-fold increase in surface roughness (Fig. 4a). The changes in roughness occur abruptly over distances as short as several hundred meters. The size of these features, labelled A, B and C in Fig. 4a, is on the order of several tens of kilometers in width. They are parallel to the main sastrugi orientation that can be seen in simultaneously collected Digital Mapping System (DMS) visual imagery and ATM spot elevation data. Figure 5 shows a close–up of one of the features (C). The CReSIS

snow radar data collected at the same time reveals the same spatial changes in surface roughness (Fig. 5 a, c). The spatially coherent segments with increased surface roughness are less pronounced in 2016 compared to 2014. The transitions from smoother to rougher surfaces are less abrupt in 2016. Both, the laser derived surface roughness and the roughness estimated from snow radar data seems to be even slightly lower than the smooth area. In 2017, the segments labelled A, B and C appear to have no distinct roughness anomaly (Fig. 4d) compared to the surrounding areas.
**Figure 5:** Laser spot elevation measurements and DMS imagery over roughness feature "C" (Fig. 4b). The center of all map panels is at 88.97° S/135° E. All panels cover the exact same area on the ground and are shown in a local coordinate system parallel to the aircraft trajectory. Panels a), c) and e) show laser spot elevation measurements in 2014, 2016 and 2017. Inset plots in a) and c) show surface roughness from ATM and snow radar at the center of the scan/nadir position. Panels b), d) and f) show corresponding DMS imagery (b) and d) only) and surface roughness from laser altimetry in 2014, 2016, and 2016. The darkening towards the edge of the DMS frames in b) and d) is caused by vignetting from the lens and not related to geophysical changes. ATM roughness is from ICESSN data and the UAF Riegl data from 2017 has been calculated in the same way as ATM ICESSN data to make them compatible. A distinct change in roughness can be seen in 2014 that is visible in both, the laser derived surface roughness, and the length of the shadows in DMS imagery (b). The roughness doubles over a distance of ~200 m (a). The orientation of the boundary is parallel to the dominant sastrugi orientation (b). In 2016 and 2017 the distinct change in roughness seems to have been smoothened out.

## 4 Snow accumulation rates derived from snow radar data and MERRA-2

For snow-radar derived accumulation calculations, we first stack traces to an approximate along-track separation of 100 meters (~ 18 traces), which largely reduces noise in the return power especially at depth. These stacked echograms are then combined into segments of ~100 km. A single radar reflection horizon, assumed isochronous, is tracked through the 100–km segment. The actual horizon picked will likely vary from segment to segment because we chose to map the strongest and most continuous reflection within that segment. The horizon is picked semi–automatically. First, the user visually selects the horizon of interest. The range





bin with the strongest return power within a 15 bin window is then selected as the horizon "pick" for that trace. That pick is extended laterally across all traces by finding the strongest return power in adjacent traces within the 15 bin window. The user can then modify the picks if they deviate from their visual interpretation. The user can also eliminate portions of a given horizon if

visual inspection deems horizon differentiation impossible. While a single, continuous horizon around the entirety of 88° S would be ideal to calculate temporally consistent accumulation rates between segments, because of the strong spatial variability in accumulation rate as well as the strength of the return, it is effectively impossible. Thus, the accumulation rates estimated for each 100-km segment will span differing time intervals; however, because of the relatively low accumulation rates, the majority span several decades minimizing the impact of interannual variability.

For each 100–km segment, we estimate the spatial variability in snow accumulation using the aforementioned horizon picks. Typically, radar derived accumulation rates rely on knowledge of the horizon age as well (e.g., Medley et al., 2013; 2014), but a lack of nearby dated ice–core stratigraphy or clearly defined annual horizons restricts our ability to assign an age to our horizon picks. Because our work is focused on evaluating the spatial variability in snow accumulation, we develop a method that approximates the age of a given horizon through combination of horizon depths and MERRA-2 mean annual precipitation–minus–

evaporation (*P-E*). In such a manner, our large–scale mean accumulation rates are forced to large–scale MERRA-2 *P-E*, however, rates are allowed to vary on < 1 km length–scales from our radar horizon picks. We detail the methodology below.

Assuming each horizon pick within a given segment is isochronous, we need to determine a way to approximate the age of that horizon. To do so, we begin by determining the mean accumulation rate and 2–meter air temperature from MERRA-2 over the entire segment, and use those variables to model steady–state firn density and age profiles using Herron & Langway (1980). The

two–way travel time ($\tau$) between the surface and the horizon pick is converted into depth (*d*) assuming

$$d(x) = \frac{c\tau(x)}{2\sqrt{\varepsilon}}, \tag{1}$$

where *c* is the speed of light and $\varepsilon$ is the integrated dielectric permittivity of the material above the horizon and *x* is the distance along flight line. Specifically, we use the modelled depth–density profile to generate depth–dielectric permittivity based on Kovacs et al. (1995), which is then used to relate depth and two–way travel time with a depth–varying radar–wave velocity. Using this

model, we interpolate horizon two–way travel time to depth. Depths vary along–track (as our layer pick varies), but our large–scale depth–age model from Herron & Langway (1980) does not; thus, we will estimate a variable along–track age of the radar horizon. The use of a single firn density model for the entire 88° S circle is justified because the difference between the minimum and maximum accumulation rates is relatively small. The along–track variability is counter to our initial assumption that the radar horizons are isochronous; however, when we take the average age along–track, we effectively force the overall mean accumulation

rate for the segment to the large–scale MERRA-2 mean. We then use this age to calculate spatially varying accumulation rates along the entire segment as outlined by Medley et al. (2015). In such a manner, we force the large–scale mean accumulation rates to those prescribed by MERRA-2 but allow for small–scale variability derived from the snow radar horizon picks in the absence of independent estimates of firn depth–age profiles (Dattler et al., 2019).

## 5 Spatial variability in snow accumulation rates

Accumulation of snow on the Antarctic ice sheet is primarily the result of precipitation of snow. The precipitated distribution of accumulated snow is subsequently modified spatially by wind–driven erosion and deposition. Sublimation of accumulated snow, both, in the form of wind-driven sublimation of airborne snow particles and surface sublimation removes accumulated snow and therefore mass from the surface and further modifies the initial deposition pattern resulting from precipitation (e.g., Frezzotti et al., 2007, and references herein). For slopes ≥ 0.002 Das et al. (2013) found wind–scoured areas in East Antarctica with negative



surface mass balance similar to the wind–glaze area described by Scambos et al. (2012). Therefore, there is no simple relationship between surface slope, wind direction and snow accumulation rates. Previous work used the Mean Slope in the mean Wind Direction (MSWD) for studying relationships between surface slope and spatial variability in snow accumulation rates (e.g., Das et al., 2013; Dattler et al., 2019; Scambos et al., 2012). MSWD is defined as the scalar dot product between the surface slope with the mean wind direction (Scambos et al., 2012). Here, we use the time-averaged zonal and meridional wind components $u$ and $v$

from MERRA-2, transformed in to a Cartesian polar-stereographic projection, to calculate the mean wind direction. Analyzing relationships between surface slope, accumulation rates, and mean wind direction at 88° S is limited by the latitudinal resolution of the MERRA-2 reanalysis model, which is 0.5° or 55 km, as well as the cross-sectional nature of the geophysical surveys (i.e., the data represent a 2-dimensional cross section). Given the narrow swath width of the ATM laser data (240 m) we use the ice surface slope derived from the CryoSat-2 DEM (Helm et al., 2014) at 1 km resolution to calculate the MSWD. The slope south of

88° S is only weakly constrained due to the absence of elevation data imposing further limitations on the analysis. The MERRA-2 26 year 10 m wind field is interpolated to the CryoSat-2 DEM grid cell locations. The difference in spatial resolution between the surface DEM and MERRA-2 will result in MSWD uncertainty from oversampling the wind field. Because of the small slopes in the study area, however, we don't anticipate complex wind fields where actual wind orientation would significantly deviate from the MERRA-2 reanalysis model. Small topographic features, however, are not represented by the 10 m surface wind field as will

be discussed later.



**Figure 6: a) Ice surface elevation around 88° S from ATM laser altimetry. b) Surface roughness from ATM laser altimetry. c) Snow accumulation rate derived from 2014 snow radar data and tied to MERRA-2. A 14 minute disk failure of the snow and Ku-band radars in 2014 resulted in a data gap over the megadunes and therefore the accumulation rate. d) Mean Slope in the mean Wind Direction (MSWD) in one degree longitude bins (4 km). e) Standard deviation σ of the MSWD and snow accumulation rate estimated over 20 km long segments.**

In general, annual snow accumulation is on the order of several cm w.e. yr$^{-1}$ and is highest near the backside of the Transantarctic Mountains near 150° W, a region that is influenced by precipitation from cyclonic events penetrating the area from the Bellingshausen and Ross Sea sectors (Casey et al., 2014; Nicolas and Bromwich, 2011) (Fig. 6). The highest accumulation rates near 150° W coincide with a megadune field (Fig. 3b) and appear to be in a local topographic low at the flank of Titan Dome that trends perpendicular to the aerogeophysical survey profile around 88° S (Fig. 6A and Fig. A1a). The surface depression coincides with a 1000 m deep and 25 km wide bedrock low perpendicular to the profile ((Studinger et al., 2006), and Fig. 3a). The dominant wind direction is near perpendicular to the survey profile and follows the trend of the ice surface low (Fig. 3a). The high accumulation area also shows high surface roughness combined with steep slopes (Fig. 4 and Fig. A1). Using European Centre for



Medium-Range Weather Forecasts (ECMWF) reanalysis data, Casey et al. (2014) estimate that around half of the snow accumulation at the South Pole comes from periodic moisture–bearing storms traversing the Filchner–Ronne and Ross Ice Shelves towards the pole from West Antarctica. It is likely that the percentage of snow accumulation from such cyclonic events is even higher at 88° S compared to the South Pole since the area is closer to the source of moisture, complicating the relationship between surface slope, wind direction and snow accumulation. Another area with relatively high snow accumulation rates is located between

45° W and 0° longitude and is also exposed to precipitation from the Weddell Sea sector (e.g., Casey et al., 2014).

The snow accumulation rate at 88° S is spatially highly variable over very short length scales of several kilometers (Fig. 6c). Small–scale variability in snow accumulation rate correlates with small–scale variability in ice surface elevation (Fig. 6a), suggesting that wind–driven erosion and deposition is a primary process of snow accumulation. The relatively constant surface elevation between 60° E and 150° E shows very little variation in snow accumulation. In contrast, the short scale (< 10 km)

undulations in ice surface elevation between 75° W and 45° E correspond to a highly variable snow accumulation pattern with similar length scale (Fig. 6 a, c and Fig. A2). The dominant wind direction in the western segment is subparallel to the profile (Fig. A2). Here, several pronounced peaks in snow accumulation rate correspond to topographic depressions in ice surface elevation (grey dashed lines in Fig. A2) indicating windblown deposition of snow. The eastern part of the profile has wind direction oblique or perpendicular to the profile. However, still several peaks in snow accumulation rate correlate with topographic depressions.

Near 90° E the wind direction is parallel to the profile (Fig. 3). A pronounced peak in snow accumulation rate at 90° E correlates with 20 m deep depression in surface topography that is several kilometers wide (Fig. A3). Accumulation decreases on the lee side of the topography high at the western shoulder of the depression and increases towards the lowest part of the depression where it reaches is highest point (Fig. A3). The general correlation of highs in accumulation rate with lows in topography results in a negative correlation coefficient of $r^2 = -0.33$ between accumulation rate and MSWD (Fig. 7a). DMS natural color imagery and

laser altimetry data shows typically two dominant wind directions (Fig. 2). Our MERRA-2 wind direction is an average over seasonal variations in wind speed and likely reflects a wind direction somewhere between the two dominant orientations of sastrugi. Since we have no knowledge of when a particular layer of snow has been deposited during a year it is not possible to do a more detailed analysis.

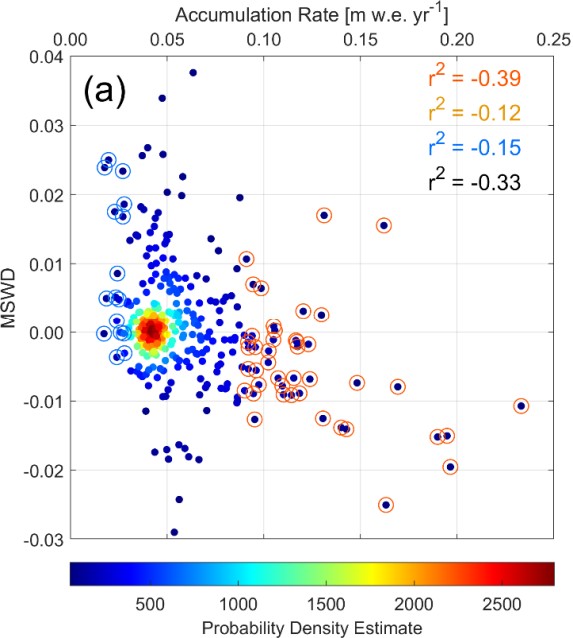

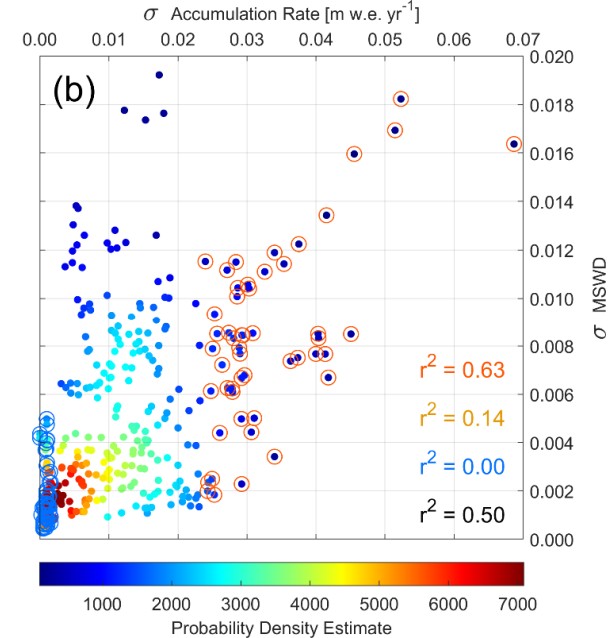



**Figure 7: a) Mean Slope in the mean Wind Direction (MSWD) versus snow accumulation rate around 88° S in one degree longitude bins (4 km). Since monochrome scatter plots can be misleading we have used color coding to indicate regions with higher point density. Higher probability density estimates, calculated using a kernel estimate, are shown in warmer colors and indicate regions with higher point density. The probability density estimate is for visual clarity only and is not used for analysis or interpretation. Pearson's correlation coefficient $r^2$ = -0.33 for the entire data set. b) Same for the standard deviation σ of the MSWD and snow accumulation rate with $r^2$ =**
**0.50. σ is estimated over 20-km-long moving windows. Data points below μ - σ are indicated with blue circles and $r^2$s are listed in blue. The upper subset consists of data points above μ + σ and is marked by red circles and red $r^2$ values. The remaining data points fall within ±σ from μ with $r^2$ shown in orange (data points are not marked for clarity).**

The surface roughness derived from ATM laser altimetry reflects roughness on a scale of <250 m and does not reflect ice surface slope changes on length scales of several km. However, the MSWD (Fig. 6d) shows the same pattern of high variability between

75° W and 45° E and fairly constant values between 60° E and 150° E. To quantify the relationship between variability in surface slope, wind direction and accumulation rates we use the standard deviation σ of the MSWD and snow accumulation rate calculated over a 20 km long moving window (Fig. 6e). Figure 6e shows the standard deviation of the MSWD and accumulation rate. In general, higher σ in accumulation rate generally occurs in areas with higher σ in MSWD. The correlation is strongest near 90° E where wind orientation is parallel to the profile.

The correlation coefficient between the standard deviations of the accumulation rate and the MSWD is $r^2$ = 0.50 indicating a stronger link between these variables than the actual parameters (Fig 7b). The magnitude of the correlation coefficient, however, is dependent on the length scale used to calculate the standard deviation. Dattler et al. (2019) find a similar behavior between σ accumulation rate and σ MSWD. Visual inspection of Fig. 7 suggests that the relationship between accumulation rate and MSWD and the σ of accumulation rate and σ of MSWD is more pronounced for larger magnitudes of the variables. A kernel density

estimate quantifies the probability density estimate of nearby points and allows visualization of the point density using a color scale for the data points (Fig. 7). We divide the data set into 3 subsets using the mean μ and σ: the lower end is defined by values below μ - σ, while the upper end are values above μ + σ. The remaining points that are within ±σ from μ form the center subset. We have calculated correlation coefficients $r^2$ for all subsets. In general, the correlation is strongest for the upper subsets, while the lower subsets show weak correlation. This is different from Dattler's et al. results (2019) who finds that the lower end also

shows strong correlation. The upper tenth percentile of our data has an $r^2$ of 0.85 similar to Dattler's et al. results (2019). Most of Dattler's et al. data (2019) is located over high accumulation areas in West Antarctica. A possible explanation for the weak correlation could be the very low accumulation rates in our area, combined with very small slopes and low wind speed. Noise in the elevation data will have a stronger impact on MSWD calculation and similarly, subtle changes in surface slope are likely below the resolution of MERRA-2, therefore resulting in a noisier and thus uncorrelated lower subset.

**6 Relationship between surface roughness, slope and wind direction**

Wind–related deposition and ablation processes could cause spatial roughness variations depending on surface slope and wind direction. For example, windblown deposition of snow into concave surface depressions and ablation on up–slope areas could create spatial surface roughness patterns that correlate with slope and wind direction. We use the MSWD to determine if slopes that are exposed to uphill winds have different surface roughness than slopes experiencing primarily downhill winds (Fig. 8a). We

calculate $r^2$ for up–slope winds (MSWD < 0) and down–slope winds (MSWD > 0). Neither the up–slope winds ($r^2$ = -0.14) nor down–slope winds ($r^2$ = 0.29) show any statistically significant correlation between surface roughness and slope as can be seen in the scatter plot (Fig. 8a). Similarly, the surface roughness does not seem to be correlated with snow accumulation rates ($r^2$ = 0.28) indicating that there is also no statistically significant slope–independent relationship between surface roughness and accumulation rates within our survey area (Fig. 8b). Correlations may exist in smaller local areas, but our data shows that there is no consistent

relationship between surface roughness, slope and wind direction on a regional scale within our survey area. However, our analysis





is constrained by using 2-dimensional high-resolution roughness estimates and correlating it with 3-dimensional wind fields and surface slope with much lower spatial resolution.

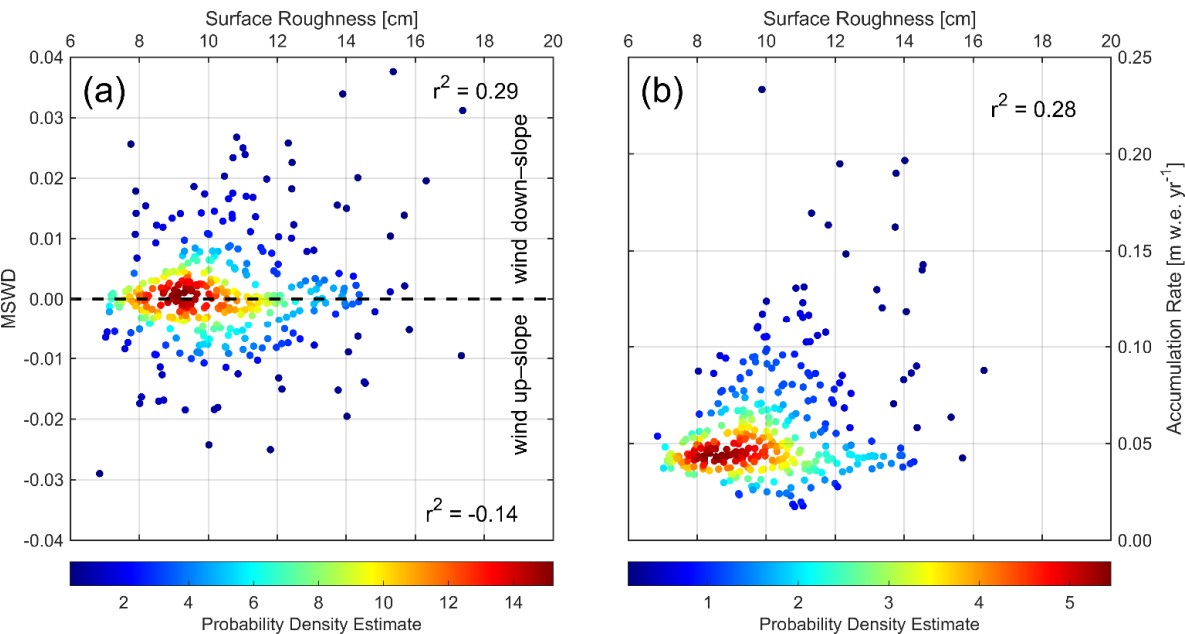

**Figure 8: a) ATM derived surface roughness versus Mean Slope in the mean Wind Direction (MSWD) around 88° S in one degree longitude bins (4 km). Color indicates the probability density estimate of nearby points using a kernel density estimate. Pearson's correlation coefficients r² are calculated for up–slope winds (MSWD < 0) and down–slope winds (MSWD > 0). b) Same for surface roughness versus accumulation rate.**

## 7 Radar backscatter and surface roughness

Surface roughness impacts the return signal of radar altimeters and can therefore cause elevation biases (e.g.,van der Veen et al., 2009, and references herein) similar to slope–dependent errors in altimetry data (Helm et al., 2014; Slater et al., 2018). Radar backscatter in radar altimeters such as ESA's CryoSat-2 is a function of surface roughness. Surface roughness at the length–scales of the radar wavelength (2.2 cm) predominantly contributes to radar backscatter which cannot be resolved by our laser data. Changes in surface roughness cause changes in radar backscatter through changes of the echo waveform which can introduce range

biases in the retrieval of surface elevation (Arthern et al., 2001; Kurtz et al., 2014). Figure 9 shows the maximum of the Ku-band radar return signal strength over the distinct roughness features identified from laser altimetry data (see Section 3.5). The maximum return energy over rougher surface areas is about 3 dB lower than over the smooth areas in between. The difference in return signal strength is even more pronounced for the snow radar (4 dB, not shown). Stacked Ku–band waveforms shown in Fig. A4 show 3 dB higher surface return power at 3.053 µs over a smooth surface compared to the rough surface. The amplitude of the subsurface

backscatter below 3.06 µs, however, is similar in strength over smooth and rough areas (Fig. A4). This observation is consistent with Gow's (1965) finding that heat from radiation causes crystal growth on the flanks of sastrugi, resulting in loosely bonded crystals that are prone to erosion by moderate winds (Gow, 1965). This differential sublimation–deflation driven redistribution of snow which flattens the surface topography at the end of the summer resulting in relatively flat subsurface stratigraphy compared to the surface topography. The difference in waveform shapes between smooth and rough surfaces suggests that radar altimeters

are potentially prone to elevation errors when threshold or leading edge trackers are being used for range retrieval. Due to the

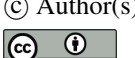



relatively wide bandwidth and small footprint size of the Ku–band radar the stacked returns in Fig. A4 allows resolution of the surface and sub-surface layers and thus accurate tracking of the surface elevation. However, the reduced bandwidth and significantly larger footprint size of CryoSat-2 LRM returns does not allow for the resolution of individual layers, but instead leads to a pronounced broadening of the return waveform when the backscatter of the sub–surface layers is close to, or exceeds, the

backscatter from the surface layer.

The relatively small–scale nature and temporal variability of these features would require the use of more sophisticated retrieval techniques to better account for differences caused by the lower relative surface backscatter of rough areas. The elevation biases caused by temporal and spatial variability in surface roughness are in addition to elevation biases caused by wind–induced anisotropy in the firn that have been identified from cross–over analysis (Armitage et al., 2014).


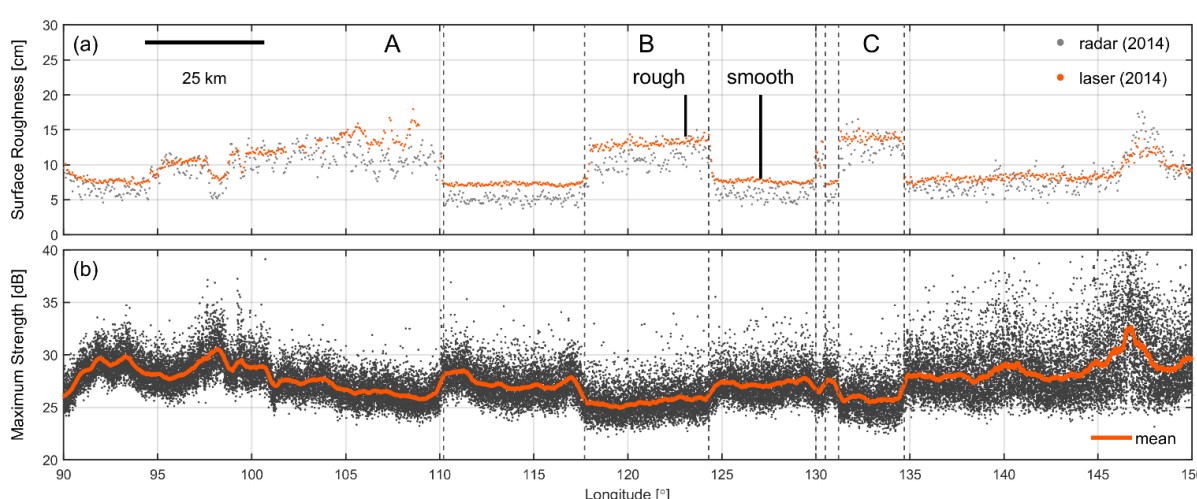

**Figure 9: a) Surface roughness from 2014 ATM laser and CReSIS Ku-band radar data over roughness features "A", "B", and "C" (Fig. 4b). Vertical black lines mark he locations of radar waveforms over smooth and rough surfaces shown in Fig. A4. b) Maximum of relative return signal strength from 2014 snow radar data. Red line is a running mean calculated over 350 radar traces (~2 km), which is similar**
**in size to the 1.65 km CryoSat-2 footprint in low resolution mode (LRM) over smooth surfaces (Scagliola, 2013). The strength of the surface return is around 3 dB weaker over rough areas compared to smooth areas.**

## 8 Conclusions

We have mapped the spatial and temporal variability in surface roughness and snow accumulation rate on a regional scale along a 1400 km circle around 88° S. We find significant small–scale variability (< 10 km) in snow accumulation based on snow radar
subsurface stratigraphy, indicating areas of strong wind redistribution are prevalent at 88° S. The observed small–scale variability in snow accumulation rates is not captured by existing reanalysis models such as MERRA-2, which suffer in spatial resolution. Our analysis shows that there is no simple relationship between surface slope, wind direction and snow accumulation rates for the entire survey area. Previous studies have primarily focused on smaller regions often showing good correlation between surface slope and accumulation rates and are often used to falsely extrapolate parameters and relationships to larger regions beyond the
study area. While we also observe these local correlations between surface slope, wind direction and accumulation rates, our results show that even for a homogenous area like the East Antarctic plateau near the South Pole such simple relationships don't exist on a regional scale. At the same time, we note that our accumulation rate measurements are a simple 2–dimensional view; until we have 3–dimensional mapping of accumulation rates, these relationships might remain elusive. Our results underline the importance of regional–scale studies to derive accurate regional–scale parameterizations and relationships in light of expanding data sets,



advances in high–performance computing and sophistication in model development. Similarly, we find high variability in surface roughness derived from laser altimetry measurements on length–scales smaller than 10 km, sometimes with very distinct and sharp transitions. These areas also show significant temporal variability over the course of the 3 survey years. We also find that surface roughness does not seem to be correlated with snow accumulation rates. There seems to be no statistically significant slope–independent relationship between surface roughness and accumulation rates within our survey area. The observed small–scale temporal and spatial variability in surface roughness will make it difficult to develop elevation bias corrections for radar altimeter retrieval algorithms.

***Data availability.*** All NASA Operation IceBridge data used in this study are freely available at the National Snow and Ice Data Center (NSIDC) at https://nsidc.org/icebridge/portal (accessed 2019). The CryoSat-2 DEM from Helm et al. (2014) is available at https://doi.pangaea.de/10.1594/PANGAEA.831392 (accessed 2019). The MODIS Mosaic of Antarctica (MOA, (Haran et al., 2014)) is available from NSIDC at https://nsidc.org/data/nsidc-0730 (accessed 2019). MERRA-2 data is available at https://gmao.gsfc.nasa.gov/reanalysis/MERRA-2/data_access/ (accessed 2018).

***Author contributions.*** MS led the analysis of the laser altimetry, optical imagery, integration of results and prepared the manuscript with contributions from all co-authors. BM derived surface roughness and accumulation rates from snow radar data and MERRA-2 and wrote the corresponding manuscript sections. KB, KC, and TN contributed to the analysis and interpretation of surface roughness and accumulation rates. NK and TO contributed to the interpretation of radar backscatter and surface roughness. SM contributed to the analysis and interpretation of the ATM laser altimetry data. All authors helped write the paper.

***Competing interests.*** The authors declare that they have no conflict of interests.

***Acknowledgements.*** We thank the Operation IceBridge instrument teams and flight crews for 11 years of data collection that made this study possible. Richard Cullather is thanked for discussions about MERRA-2. Mark Fahnestock is thanked for discussions about surface roughness, slope and accumulation rates.

***Financial support.*** Funding for this work comes from NASA's Cryospheric Sciences Program.

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





# Appendix A


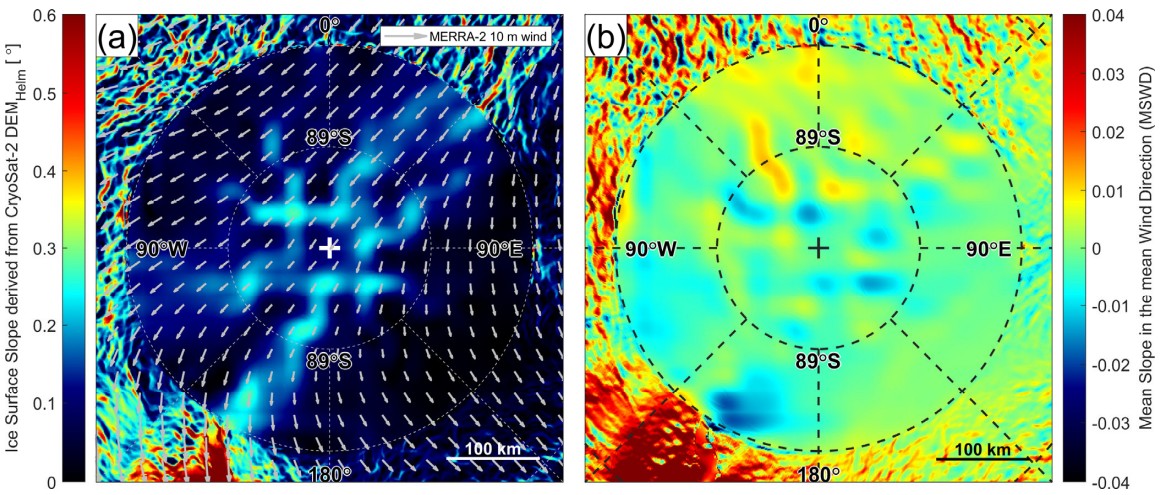

**Figure A1: a) Ice surface slope in degrees derived from CryoSat-2 (Helm et al., 2014) and 26 year 10 m wind average from MERRA-2**
**(e.g., Gelaro et al., 2017). b) Mean Slope in the mean Wind Direction (MSWD).**





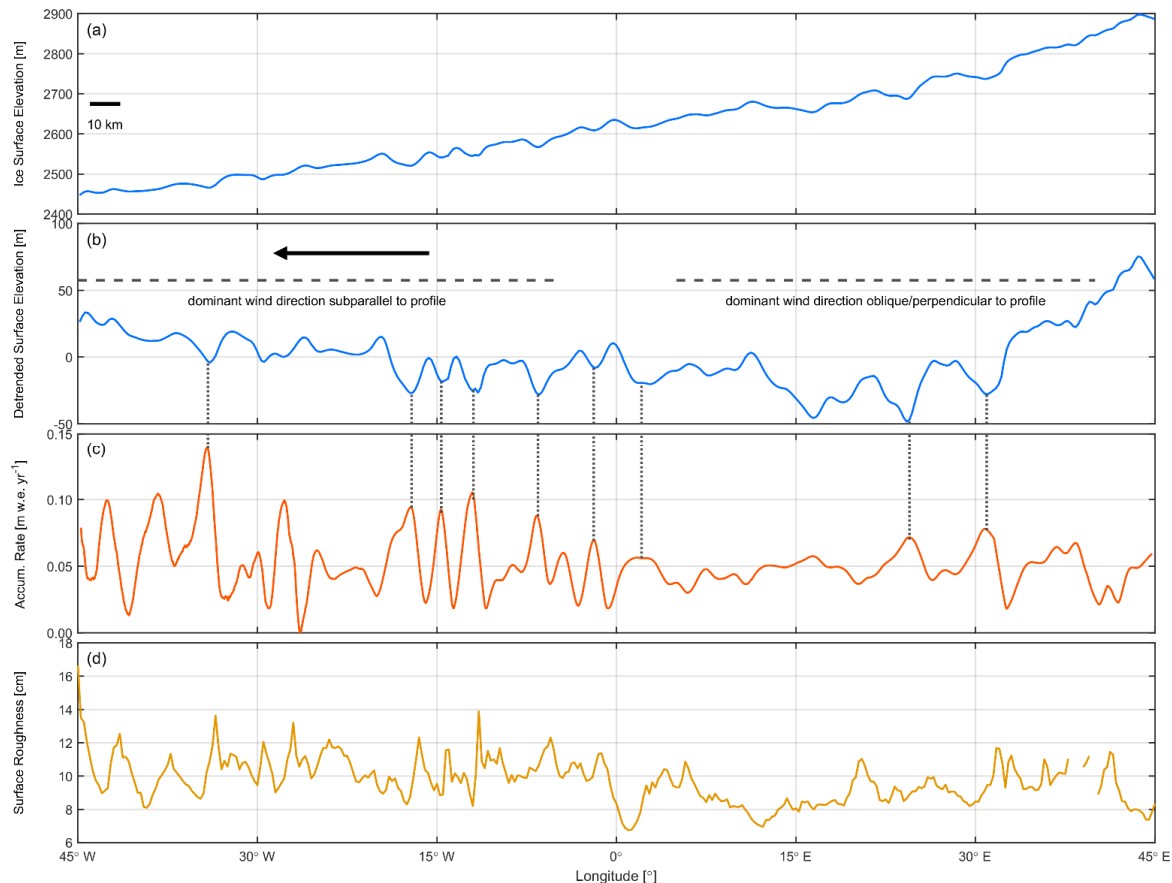

**Figure A2: a) Ice surface elevation at 88° S between 45° W and 45° E from ATM laser altimetry. b) Ice surface elevation with linear trend removed. c) Snow accumulation rate. Several prominent peaks that spatially correlate with topographic depressions are marked**
**by grey dashed lines. d) ATM derived ice surface roughness.**




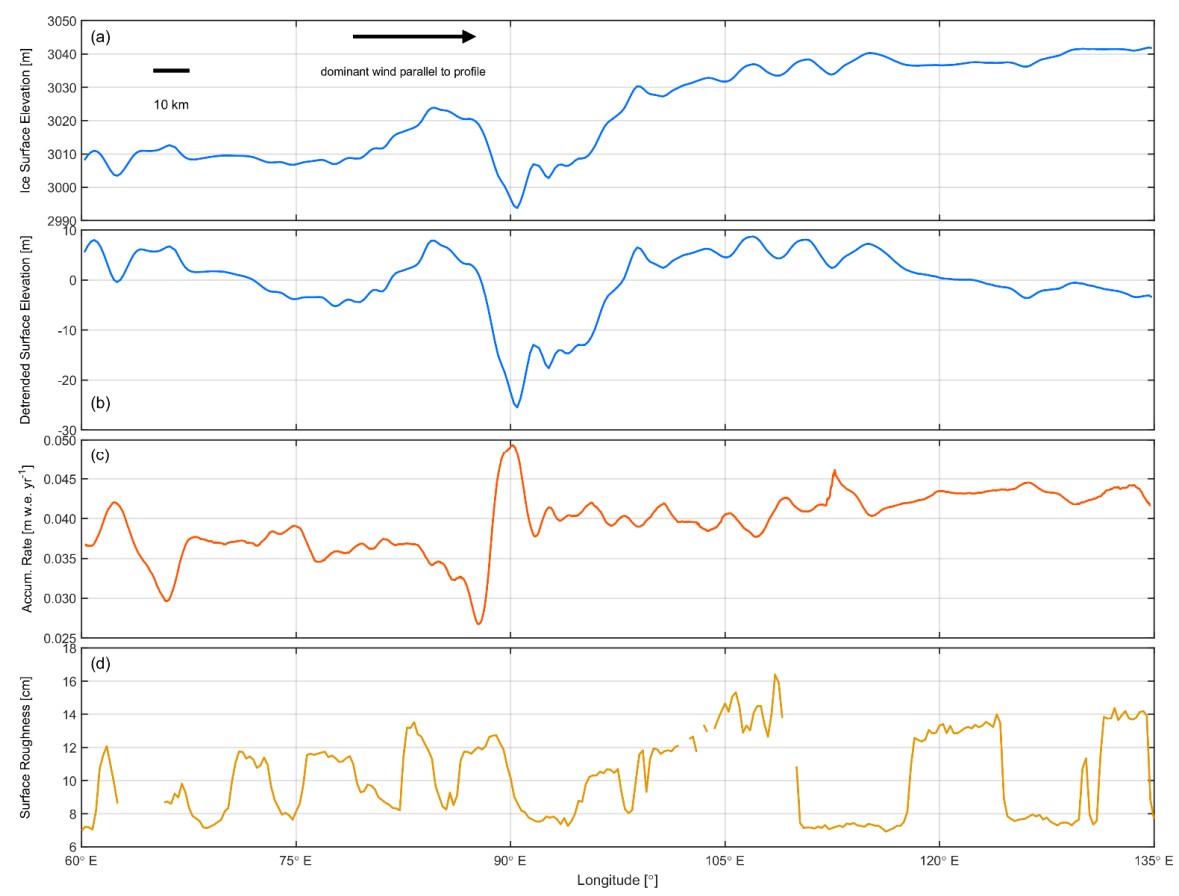

**Figure A3: a) Ice surface elevation between 60° E and 135°E from ATM laser altimetry. The dominant wind direction from MERRA-2 is parallel to the profile segment. b) Ice surface elevation with linear trend removed. c) Snow accumulation rate were the pronounced**

**peak at 90° E corresponds to the topographic depression in the ice surface. d) ATM derived ice surface roughness.**

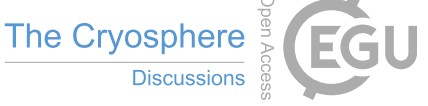



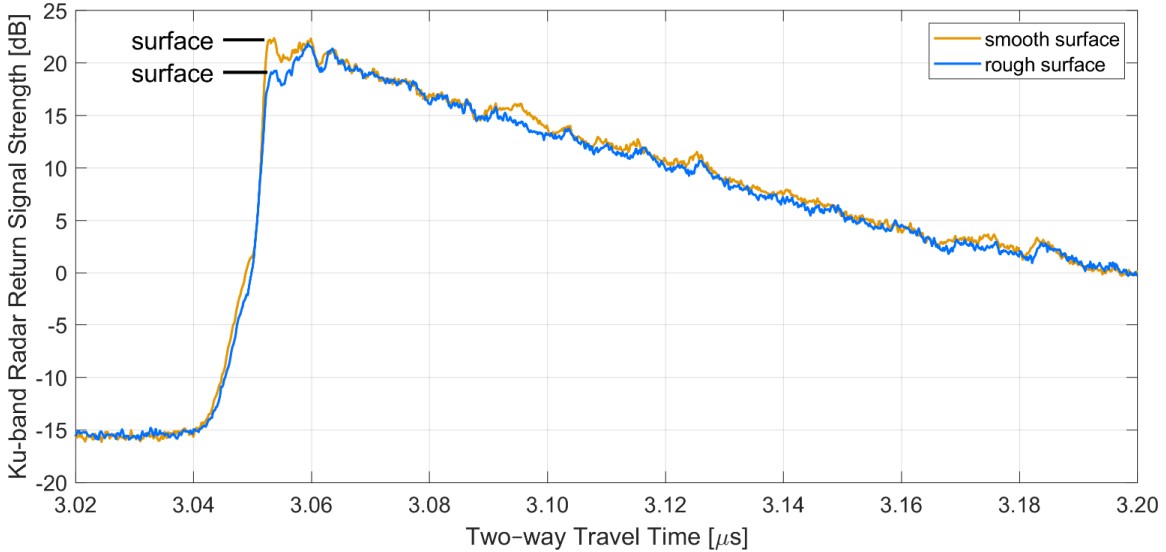

**Figure A4: Relative Ku-band radar return signal strength over smooth and rough surfaces. 100 traces were stacked for the averaged waveforms. For location see Fig. 9. The difference between the radar waveforms indicates that radar altimeters are potentially prone to elevation errors when threshold or leading edge trackers are being used for range retrieval.**
