# Peer review of "Temporal and spatial variability in surface roughness and accumulation rate around 88° S from repeat airborne geophysical surveys"

_The Cryosphere, 2020_

## Referee Comment (RC1) · Joshua Chambers (Referee) · 23 Apr 2020

General comments:

The authors of this well-presented study make good use of existing, open datasets to address an important aspect of remote sensing. Centred around the validation of 2 contemporary remote sensing platforms widely used by others, the study site and datasets place this article well within the remit of the journal. Multiple sources of geophysical data remotely sensed from airborne platforms are used to investigate surface roughness, slope and wind direction along the whole latitude of 88°S, along with a novel way to model accumulation rate.

[Figure]

The use of MSWD is here justified through use in other studies, although it seems as though the very low surface gradient and low accumulation rates would make it difficult to find a balance of scales that would show a clear relationship. Low accumulation rates and relatively low amplitude roughness at the scales studied present a similar issue. As a suggestion for further possible investigation (and being aware that the authors acknowledg a lack of 3D data), I wonder if surface anisotropy would show any correlation with accumulation? Particularly as sastrugi are widespread and have a directional element.

Overall the work was structured logically, clearly articulated and would be reproducable. The methods are well explained and justifiable, with some reasonably thorough interrogation of the findings. I think this is a worthwhile contribution despite the overall negative outcome. The authors are aware of the limitations of the data used and are frank about not being able to achieve the initial aim. Importantly, they don't try to overstate their findings. I would recommend the study be published with the minor corrections suggested here, along with any suggestions from other reviewers.

Specific comments:

Lines 94-94 – what is the normal annual layer thickness? An example and a citation here would be helpful

Line 106 – insert (sastrugi) after "distinct elongated snow surface features" and delete next sentence on line 107.

Lines 131, 160, 182 & 185, 324 – 'herein' should be 'therein'

Line 134 – no need to define sastrugi again. This sentence can be merged with the following sentence.

Line 160 – Merge Gow (1965) citation into previous citation, separated by a semicolon.

Line 165 – change "Slope-dependent accumulation are" to "Slope-dependent accumulation is"

Line 188 & 191 – I guess by '3rd order polygon' you mean 3rd order polynomial?

Line 189-191 – did you test the different methods of detrending? If so, I think including a brief summary in the supplementary information would be appropriate; if not, I'd change the language ('significant difference' implies a statistical test was applied) and cite a study that has shown this to be the case.

Line 192-194 – you refer to range bins in the sentence before defining them in the next sentence. These two sentences can be merged.

Line 198 – how many points were discarded? Maybe include a value or percentage here

Line 201 – include velocity value here for reader-reference

Line 203 – ICESSN doesn't seem to be defined anywhere

Line 235 – is the reason for discounting ice-dynamics related roughness just to be sure that features interpreted as sastrugi are sastrugi, for mapping wind direction? Otherwise, why make the distinction?

Line 252-253 – if the dunes move ∼60m in 34 years, would the ∼7 m movement in 4 years make that much difference to the coverage of the data? Also, what kind of process moves the dunes, is it aeolian or ice dynamics?

Figure 5 – do the semi-circular artefacts in panels (a) and (b) have elevation values, or are they no-data? (My eyes aren't good enough to tell!). Do they affect the roughness values at all? Why do they not appear in (e)?

Line 324 – units missing from slope value, i.e. slopes $\geq 0.002°$

Figure 6 – Slope units also missing here, assuming the correct unit for MSWD is degrees?

---

## Referee Comment (RC2) · Neil Ross (Referee) · 28 May 2020

**Review of: Temporal and spatial variability in surface roughness and accumulation rate around 88 S from repeat airborne geophysical surveys" by Michael Studinger et al. (MS No.: tc-2020-51)**

This manuscript provides a detailed and in-depth report of surface roughness and accumulation over the Antarctic Plateau. The analysis uses the airborne NASA/Operation IceBridge ICeSat-2 calibration surveys around the 88° S latitudinal limit of ICESat-2 and CryoSat-2. The authors' results demonstrate the complexity and variability of surface roughness and accumulation rates over wide areas, and the manuscript makes a substantive and important contribution to the discipline and study region.

The manuscript is very well-written, datasets and methods are comprehensively reported, and the data are presented effectively. The manuscript is full of useful and important scientific content, but I am not fully convinced that the key messages and impact of the paper are maximised, or as accessible, as they could be (see 'major' issues below). The primary finding of the manuscript is a negative result (i.e. "our analysis shows that there is no simple relationship between surface slope, wind direction and snow accumulation rates for the entire survey area"), but that is not an issue in any way. _It is imperative that such negative results are reported and published_. The main finding here has important implications for our understanding of the relationships between these factors at local (correlation) and regional (no correlation) scales.

**Broad comments:**

There are 5 issues that I believe are worth highlighting about the manuscript in its current form. Were these to be addressed, the manuscript would be much improved:

1.  A lack of discussion about the wider implications of the study. I would encourage the authors to consider the broader implications of this work (e.g. implications for satellite-derived accumulation rates, implications for ice core research at South Pole, Hercules Dome etc.) and incorporate these aspects into the abstract, discussion and conclusions sections.

2.  A lack of information about the geographical setting of the area of investigation (e.g. which Antarctic drainage basins and ice stream catchments does the 88°S survey line intersect and survey?). A 'study area' section to the manuscript may help in this regard.

3.  A somewhat awkward structure to the paper, with inter-mixing of study area, methods, results, and background information. There is no clear delineation between description of results and discussion/interpretation of those results. To me the current structure is not that logical, but this may reflect my training/background discipline. However, a decision on what section 3 is needs to be made. Is it a methods section, or something else? Currently it is an amalgamation of background, methods and results.  Section 4 also seems to be predominantly methods, with sections 5-7 results/discussion intermingled. My recommendation is to restructure the manuscript to more clearly delineate between background/methods/results/discussion & interpretation. Without this restructuring it will not be clear whether the manuscript is a 'methods' paper or a 'results' paper.

4.  A lack of citation to previous, potentially relevant, surface accumulation/roughness investigations in study area (e.g. results from the South Pole-Queen Maud Land traverses of 1964-65 and 1965-66), and potentially to Antarctic-wide surface accumulation literature (e.g. Arthern et al. 2006 https://agupubs.onlinelibrary.wiley.com/doi/10.1029/2004JD005667).

5. Statistics are not my strongest suite, so I am not terribly well qualified to comment on the validity of those aspects. The statistics used seem simple and relatively unsophisticated however (e.g. lines 391-409), and I do wonder if a more sophisticated and rigorous statistical analysis has the potential to tease out more insights from the data set. Perhaps the current statistical analysis does the job however (i.e. 'reviewer 2' is simply scaremongering unnecessarily), and there is no requirement for introducing more complexity in this regard. I was a little unsure as to why it was so important to assess the correlation of the standard deviation (lines 391-409) however. Perhaps this could be explained a little more?

**Specific comments:**

Line 23: "overall" instead of "entire"?

Lines 26-28: Implications of this study for developing elevation bias corrections is stated, but I'd encourage the authors to include other broader implications (e.g. for quantification of surface mass balance and for ice core studies) here.

Line 46: "…ice surface, volume…"?

Line 47: "Radio-wave signal detection below the noise floor…"?

Line 50: "…azimuth dependent elevation ….."?

Line 52: "…we specifically studied…."?

Line 62: Change title to "Data sets and methods"?

Line 85: no comma after "both"?

Line 86: "…a ground speed…"

Line 105-108: Change to: "The difference in geolocation between distinct elongated topographic snow surface features (sastrugi) between overlapping orthorectified images is on the order of several metres. The DMS images….."?

Lines 119-123: More detail on the survey flights would be helpful. For example, what was terrain clearance, and was it consistent each year?

Line 127: Section "3.1 Background": It is unclear to me why this section is titled background, and why it is positioned here. If the section is background, then it should probably come before datasets/methods. This section seems to be a mix of background (lines 128-137 & 150-157) and surface roughness results (137-144).

Line 134: "Elongated….sastrugi" is repetition of line 105-108.

Line 155: Use of Gow reference is really effective. Engagement with more literature of this age could benefit the paper, e.g. publications from the South Pole-Queen Maud Land traverses of 1964-65 and 1965-66. Lots of the results of these traverses are published in "Antarctic Snow and Ice Studies II" https://agupubs.onlinelibrary.wiley.com/doi/book/10.1029/AR016

Line 158: Section "3.2. Relevance of surface roughness…..": This seems to be background material rather than results.

Line 170: No need for an acronym for bidirectional reflectance distribution function. It is only used once after this in the entire manuscript.

Line 172: "the relationships"

Line 180: Section "3.3. Surface roughness estimates": The entirety of section 3.3 seems to be methods, or assessment of methods (lines 207-216) rather than results or discussion. An overall decision needs to be made about what section 3 is (see 'broad comments' above).

Line 234: year of Mouginot reference (2019?) is missing.

Line 239-249: Description of the longitudinal ranges of features of interest. I did find it tricky when reading the text to think about compass bearings in both westerly and easterly compass directions in a single sentence. I understand why the authors have done this (i.e. to describe what is shown in figures 3 and 4), but it is a little jarring and non-intuitive when reading a single sentence. For example, for me it is much easier to comprehend "between 175° W and 60 °E" when written as "between 60-185°E". As currently written, it is also not clear without reference to the figure whether the smooth area is clockwise or anti-clockwise between 175° W and 60 °E. It may also be worth considering annotating the area described in figure 3 (&4)?

Line 250: I am not sure what "…appears to have less of a roughness anomaly…." means. Why not just "..the surface of this dunefield is less rough in 2017 compared to 2014 and 2016"?

Line 253: "beyond" rather than "out of"?

Lines 254-257: this paragraph seems a little 'bolted-on' to this section and is a little perfunctory. Does it need a few more sentences to describe the data presented in figure 4e a little more fully? This section is entitled "3.4 Surface roughness, slope and elevation…" but is very much dominated by roughness.

Line 254: Why "…slope of the ATM ICESSN nadir platelets…"? Why not just "ATM-derived surface slope"?

Line 262: A, B & C are labelled in 4b, rather than 4a.

Lines 265: "simultaneously" or "concurrently" rather than "at the same time"?

Line 267: no comma after "Both"

Line 268: "…seems to be even slightly lower…" – recommend rewording to either quantify this statement, or to make it more certain. Perhaps "…are slightly lower…"?

Lines 282-318: "Section 4" – this all seems to be method description here, rather than description of data or results? Perhaps a re-structuring of the paper is required to make it clear to the reader which sections are methods, results and interpretation? If the manuscript is a methods development paper then that's fine, but that's not the impression currently given by the abstract.

Lines 320-340: "Section 5" – the first part of this section is a mix of background information (i.e. lines 320-329) and further description of methods (i.e. lines 329-340). It does not describe "Spatial variability in snow accumulation rates". I would suggest that the opening line of section 5 is not the best place to state "Accumulation of snow on the Antarctic ice sheet is primarily the result of precipitation of snow". Such a sentence should be on the 1st page of a manuscript. The entirety of Lines 320-324 should be much earlier in the manuscript.

Line 347: rather than "several", can the authors provide a range (e.g. 0-3 cm w.e. yr$^{-1}$)?

Line 349: again, here it would be good to quantify the statement made (e.g. "the highest accumulation rates (xx cm w.e. yr$^{-1}$) near…."

Line 352-353: Good to cite original paper locating the bedrock low (i.e. Studinger et al. 2006). Authors could also add an up-to-date reference here to reflect new bed data acquisition in this area. Either Paxman et al., 2019 https://agupubs.onlinelibrary.wiley.com/doi/full/10.1029/2018GC008126 or Morlighem et al., 2019 https://www.nature.com/articles/s41561-019-0510-8 ? Perhaps also move the reference to figure 3a to earlier in the sentence as it only shows surface depression, rather than the subglacial topographic low?

Line 354-360: this is an extensive description of previous work that is not directly linked into the data description/interpretation here. Could it be moved to a 'study area' section earlier in the manuscript? It might be more effective there, and can then simply be referred to at its current location?

Line 362: suggest insert an r$^2$ value after "correlates".

Line 365: "highly variable" – requires some quantification in the text (i.e. range of values should be quoted).

Line 369: Reword to "However, several peaks in snow accumulation rate still correlate…"?

Line 370: again, insert an r$^2$ value after "correlates"?

Line 372: "topographic" rather than "topography"?

Line 372-373: change to "..lowest part of the depression where it reaches it's highest point."? Perhaps quantify the "highest point" too? How high was it? Such statements should be quantified in the text.

Lines 373-374: change to "…with lows in topography results in an overall negative correlation coefficient of…."

Lines 393-394: insert an r$^2$ value after "The correlation is strongest"? I note that in-text quantification of data description is much better in the following section 6.

Line 432: is there really a requirement to say "ESA's CryoSat-2"? Why not just CryoSat-2?

Line 466: change to "….MERRA-2, which have low spatial resolutions."?

Lines 467-471: this is a very important finding.

**Figures:**

Figure 1: Cite source of rock outcrop polygons (Antarctic Digital Database?)

Figure 2: I found it difficult to orient myself between figures 2 and 3. Where is figure 2 located on figure 3?

Table 1: This is table is really useful.

**Dr Neil Ross**
**Newcastle University**
**28$^{th}$ May 2020**

---

## Author Response (AR1)

**Response to the Editor's findings (Nanna Bjørnholt Karlsson) on manuscript TC-2020-51: Temporal and spatial variability in surface roughness and accumulation rate around 88° S from repeat airborne geophysical surveys**

We use the following color and font coding scheme in our response:

**Editors's or Referee's comments**

> *Response:* authors' response to comments.

Dear Dr. Karlsson,

We would like to thank you for the many comments and suggestions that helped improve the manuscript. We have revised our manuscript accordingly. Our response to your specific suggestions are below, followed by our detailed response to RC1 and RC2's comments and the revised marked-up manuscript.

Best regards,

Michael Studinger

**Clarify in the paragraphs discussing dune migration (lines 252-253) that the rate of movement of the dunes described by Fahnestock cannot be transferred directly to your study area (e.g. different setting/area). In other words, that the cited 60 m in 34 years is just an example of what the rate might be.**

> *Response:* We have clarified the sentence accordingly: "Fahnestock et al. (2000) found 60 m of dune migration over a 34 year period for a megadune field in the vicinity of Vostok Station far away from the dune field discussed here. Therefore dune migration rates could be very different between the two sites."

**Please include a brief explanation of the artefacts in Fig 5.**

> *Response:* We have added a brief explanation to the caption of Fig. 5 and refer the reader to Yi et al. (2015) for more details: ".Panels a) and c) show small (cm level) semi-circular elevation biases that are a result of occasional variations in scan azimuth speed (Yi et al., 2015). The peak-to-peak amplitude of these biases is an order of magnitude smaller than the ice surface topography."

**I agree with reviewer 2 that a slightly longer explanation is needed to understand the reason for assessing the correlation of the standard deviation (lines 391-409).**

*Response:* We have added an explanation why we use the standard deviation of the MSWD. Also see our response to Referee 2's fifth point.

**Fig. 3/line 239: add an arrow to Fig. 3 showing where the area is located**

*Response:* We are unclear what the Editor means with this comment. This paragraph primarily describes features in Fig. 4 and is using the geographic longitudes shown on the abscissa of Fig. 4 (and in map view on Fig. 3) to identify those features. The location of Titan Dome (TD) is also marked in both figures.

In response to RC2 comments we have already marked and labelled the location of Fig. 2 in Fig. 3b and also added a remark to the caption of Fig. 2.

**lines 362-363: Since you do not calculate r2-values, please clarify if this correlation is based on visual inspection of the plots in Fig. 6, or if it is an established correlation from previous work (then cite relevant literature)**

*Response:* We have clarified the statement accordingly: "Based on visual inspection small–scale variability in snow accumulation rate correlates with small–scale variability in ice surface elevation (Fig. 6a), suggesting that wind–driven erosion and deposition is a primary process of snow accumulation."

**line 393: consider using "match" instead of "correlation" to avoid implying a statistical analysis.**

*Response:* We have replaced "correlation" with "match".

**In the response to referee 2 regarding lack of citation to previous, potentially relevant, surface accumulation/roughness investigations it is stated that: "Our paper describes spatial variations in accumulation rates and does not focus on absolute values, because there are no existing data that we could use to tie our radar-derived accumulation rates to firn cores or snow pits in the area."**

**There might not be any temporally overlapping datasets but investigating some of the literature suggested by the referee reveals that accumulation rates exist from the traverses conducted in the 1960s. For example, measurements from 1962/63 traverse (partly following 88S) showed accumulation rates of around 8g/cm2. This dataset also relates to a comment by referee 1 regarding annual layers: The 1962/63 data show 8 years are present in a depth interval of 25-190cm. See https://agupubs.onlinelibrary.wiley.com/doi/10.1029/AR016p0209 (Taylor)**

*Response:* We have plotted all existing accumulation measurements from Favier et al. (2013) that are within 10 km of 88°S in Fig. 6c. These include the 1962/63 South Pole traverse from Taylor (1971) mentioned above. We have added text at the end of Section 4 that discusses the comparison: "For comparison we have plotted all existing accumulation measurements of Favier et al. {, 2013 #109} in Fig. 6c over our MERRA-2 and radar-derived accumulation rates. These snow pit measurements include data from the 1962-1963 South Pole Traverse (Taylor, 1971). While there is general agreement it should be pointed out that Favier et al. (2013) applied the quality rating of Magand et al. (2007) , which identifies all snow pit data points shown in Fig. 6c as low

quality and subsequently excludes these data points from the quality controlled version presented in Favier et al. (2013). Further limitations of the comparison are the long time between the snow pit measurements and airborne data and the large variability in snow accumulation rates on length scales of 10 km that can be seen in the radar-derived snow accumulation rates."

**A study from the same issue of Antarctic Research Series notes that: "Detailed profiles reveal topography of the order of 10 to 30 meters in amplitude with half-wave-lengths of 10 to 30 km. Superimposed on these are features generally 2 to 4 km in extent and with up to 6 or 8 meters of relief." https://agupubs.onlinelibrary.wiley.com/doi/10.1029/AR016p0039 (Beitzel). While the amplitude is different from the numbers you report, this could be an early mention of megadunes.**

> *Response:* We thank the Editor for bringing this paper to our attention. It is unclear to us which area the above statement on page 44 of Beitzel (1971) refers to. We agree with the Editor that this is an interesting observation, however, without knowing where along the vast SPQML traverse route this observation is, we feel a speculation if this was an early observation of megadunes will not add context to our paper.

**Finally, an earlier study by Lister and Pratt (https://www.jstor.org/stable/1791117) note the dominant direction and mean height of sastrugis measured in 1957 during an expedition crossing 88S. Again, I acknowledge that the data are not temporally overlapping with your observations but a short sentence mentioning the similarity/dissimilarity with your results would be very interesting, and at the same time demonstrate the long-lived interest in surface roughness investigations.**

> *Response:* We thank the Editor for bringing this paper to our attention. We agree that this is an interesting early mention of sastrugi and have added the references and a general statement to Section 3.1. Unfortunately, the geographic area of their description is also unclear from the Lister and Pratt (1959) paper.
* * *
**Response to the Referee 1 comments (RC1 - Joshua Chambers) on manuscript TC-2020-51: Temporal and spatial variability in surface roughness and accumulation rate around 88° S from repeat airborne geophysical surveys**

*We thank referee Joshua Chambers for the positive general comments and many helpful specific suggestions. We have revised our manuscript accordingly.*

**Lines 94-94 – what is the normal annual layer thickness? An example and a citation here would be helpful**

> *Response:* There are two aspects to this question. The first one is whether the radar has sufficient resolution to resolve the layers and the second one is whether annual layers actually exist. We have reworded the sentence in lines 93-94: "At these low accumulation sites, preservation of

reflection horizons is greatly reduced due to a slow rate of burial. Also, the ambient conditions required to generate seasonal reflections might not always be present (such as depth hoar)."

**Line 106 – insert (sastrugi) after "distinct elongated snow surface features" and delete next sentence on line 107.**

*Response:* done

**Lines 131, 160, 182 & 185, 324 – 'herein' should be 'therein'**

*Response:* done

**Line 134 – no need to define sastrugi again. This sentence can be merged with the following sentence.**

*Response:* done

**Line 160 – Merge Gow (1965) citation into previous citation, separated by a semicolon.**

*Response:* done

**Line 165 – change "Slope-dependent accumulation are" to "Slope-dependent accumulation is").**

*Response:* done

**Line 188 & 191 – I guess by '3rd order polygon' you mean 3rd order polynomial?**

*Response:* We have replaced "polygon" with "polynomial" in lines 188 and 191.

**Line 189-191 – did you test the different methods of detrending? If so, I think including a brief summary in the supplementary information would be appropriate; if not, I'd change the language ('significant difference' implies a statistical test was applied) and cite a study that has shown this to be the case.**

*Response:* We have deleted the sentence.

**Line 192-194 – you refer to range bins in the sentence before defining them in the next sentence. These two sentences can be merged.**

*Response:* We have merged the two sentences.

**Line 198 – how many points were discarded? Maybe include a value or percentage here**

*Response:* 1.8% of the data points were discarded. We have added the percentage to the sentence.

**Line 201 – include velocity value here for reader-reference**

*Response:* The velocity of electromagnetic waves in air is approximately the speed of light in vacuum. We have added $2.998E^8$ m/s.

**Line 203 – ICESSN doesn't seem to be defined anywhere**

*Response:* Correct. The ICESSN format was created in the early 1990s and information for what it initially stood for has been lost. ICESSN has been known in the community for over 25 years and is now used as a name, rather than an acronym. We have rephrased line 203 to clarify that: "For

a closer look at temporal changes in surface roughness around 88° S we use the roughness estimates contained in the ATM Level 2 smoothed ice surface data product, known as ICESSN (Studinger, 2014, updated 2018)."

**Line 235 – is the reason for discounting ice-dynamics related roughness just to be sure that features interpreted as sastrugi are sastrugi, for mapping wind direction? Otherwise, why make the distinction?**

*Response:* As described in lines 129 – 134 surface roughness caused by ice flow has very different length scales compared to the features we describe here, which is the reason we are confident that these sastrugi are in fact wind-related. We reiterate this argument again in lines 234 – 236 of the initial manuscript.

**Line 252-253 – if the dunes move ~60m in 34 years, would the ~7 m movement in 4 years make that much difference to the coverage of the data? Also, what kind of process moves the dunes, is it aeolian or ice dynamics?**

*Response:* In line 251 we state "The temporal stability of megadune fields remains poorly understood." To our knowledge there is only one published dune migration rate which is the 60 m in 34 years from Fahnestock et al. (2000), which is for a mega-dune field in the vicinity of Vostok and far away from the one we describe. If the underlying ice moves, the dune field on the surface moves with the ice. But dune fields can also migrate independent from ice motion, a process that is not well understood. Given the lack of satellite imagery at 88°S we don't know what the migration rate of this particular dune field is but we have to at least consider dune migration a possibility. We have described this in lines 252 – 253 of the initial manuscript: "Since our survey area is near the edge of the dune field we cannot rule out that over the course of 4 years the edge of the dune field has migrated out of the coverage of the airborne geophysical data."

**Figure 5 – do the semi-circular artefacts in panels (a) and (b) have elevation values, or are they no-data? (My eyes aren't good enough to tell!). Do they affect the roughness values at all? Why do they not appear in (e)?**

*Response:* We assume the reviewer means instrument-related elevation biases that are a result of occasional scan azimuth biases in the ATM instrument. The effect is described in Yi et al. (2015, DOI: 10.1109/TGRS.2014.2339737). These small elevation biases become visible over extremely flat surfaces and are of the order of several cm. We do not believe that they significantly impact our roughness estimates since the elevation range that defines the surface roughness is on the order of several tens of cm. These artefacts do not appear in panel (e) because the data shown here were collected with the Riegl scanner, a linear line scanner, which is different from the conically scanning ATM instrument used for (a) and (c).

We have added a short description of the artefacts to the caption of Fig. 5 and refer the reader to Yi et al. (2015) for more detail.

**Line 324 – units missing from slope value, i.e. slopes ≥ 0.002**

*Response:* The quoted slope value from Das et al. (2013) is in meters per meter and is therefore dimensionless.

**Figure 6 – Slope units also missing here, assuming the correct unit for MSWD is degrees?**

> *Response:* The slopes are in meters per meter and are therefore dimensionless.
* * *
**Response to the Referee 2 comments (RC2 – Neil Ross) on manuscript TC-2020-51: Temporal and spatial variability in surface roughness and accumulation rate around 88° S from repeat airborne geophysical surveys**

*We thank referee Neil Ross for the positive general comments and the many and very helpful specific suggestions. We have revised our manuscript accordingly.*

**Broad comments:**

**1. A lack of discussion about the wider implications of the study. I would encourage the authors to consider the broader implications of this work (e.g. implications for satellite-derived accumulation rates, implications for ice core research at South Pole, Hercules Dome etc.) and incorporate these aspects into the abstract, discussion and conclusions sections.**

> *Response:* Section 3.2 "Relevance of surface roughness and slope for altimetry and surface mass balance" already addresses the wider implications of our work. Since we don't present specific results in our paper other than for radar altimetry we feel any statements on potential impacts on other areas of research we could make will be speculative and don't belong into a scientific paper.

**2. A lack of information about the geographical setting of the area of investigation (e.g. which Antarctic drainage basins and ice stream catchments does the 88°S survey line intersect and survey?). A 'study area' section to the manuscript may help in this regard.**

> *Response:* We have added a survey area section to the manuscript and moved some of the text in sections 3 and 4 into the new section. We have tried and plotted the drainage basins in Fig. 1 but feel the basin outlines are too distracting and don't add much context to the figure. Instead we have described the geophysical setting. Given the low ice surface velocities in our survey area (< 10 m yr$^{-1}$) we don't think drainage basins are relevant for the work presented in the paper.

**3. A somewhat awkward structure to the paper, with inter-mixing of study area, methods, results, and background information. There is no clear delineation between description of results and discussion/interpretation of those results. To me the current structure is not that logical, but this may reflect my training/background discipline. However, a decision on what section 3 is needs to be made. Is it a methods section, or something else? Currently it is an amalgamation of background, methods and results. Section 4 also seems to be predominantly methods, with sections 5-7 results/discussion intermingled. My recommendation is to restructure the manuscript to more clearly delineate between background/methods/results/discussion & interpretation. Without this restructuring it will not be clear whether the manuscript is a 'methods' paper or a 'results' paper.**

*Response:* We agree with the referee that the structure of the manuscript can be clarified. Our paper is a synthesis of several different data types (laser altimetry, optical imagery, subsurface radar) that explores relationships between several geophysical parameters that can be derived from these data. The nature of our analysis complex and this complexity is reflected in the manuscript. In order to help the reader understand the organizational structure and flow of our manuscript we have added several sentences at the end of the introduction that describe the outline of our paper. We have change the title of several sections to better reflect their contents. We have chosen to keep background, method, results and discussions together with roughness and accumulation sections to avoid the paper becoming repetitive.

**4. A lack of citation to previous, potentially relevant, surface accumulation/roughness investigations in study area (e.g. results from the South Pole-Queen Maud Land traverses of 1964-65 and 1965-66), and potentially to Antarctic-wide surface accumulation literature (e.g. Arthern et al. 2006 https://agupubs.onlinelibrary.wiley.com/doi/10.1029/2004JD005667).**

*Response:* We have provided a detailed response to this comment in our response to the Editor. We have plotted all existing accumulation measurements from Favier et al. (2013) that are within 10 km of 88°S in Fig. 6c.

There seems to be general agreement in the surface mass balance community that re-analysis models such as RACMO2.3 and MERRA-2 are now better than the Arthern et al. (2006) accumulation estimates. This is the reason why we have chosen to use a re-analysis model over the Arthern et al. (2006) data. We use MERRA-2 since unlike RACMO2.3 it is publicly available. We have added a reference to Arthern et al. (2006) in the survey area section but are not convinced of its relevance to our work. We have also added a reference to the surface mass balance data set from Favier et al. (2013). An updated and quality controlled surface mass balance dataset for Antarctica. Cryosphere, 7(2), 583-597.

**5. Statistics are not my strongest suite, so I am not terribly well qualified to comment on the validity of those aspects. The statistics used seem simple and relatively unsophisticated however (e.g. lines 391-409), and I do wonder if a more sophisticated and rigorous statistical analysis has the potential to tease out more insights from the data set. Perhaps the current statistical analysis does the job however (i.e. 'reviewer 2' is simply scaremongering unnecessarily), and there is no requirement for introducing more complexity in this regard. I was a little unsure as to why it was so important to assess the correlation of the standard deviation (lines 391-409) however. Perhaps this could be explained a little more?**

*Response:* This is an excellent point that the authors have discussed at length and with colleagues while doing the analysis. We don't think that more sophisticated or rigorous statistical methods would reveal any meaningful relationships in the data. The Pearson's correlation coefficient may not be the most sophisticated method but it is widely accepted and certainly robust, which is the reason why we decided to use it as a metric. Using the standard deviation for quantifying variability is a commonly used method. We have explained in lines 390 – 391: "To quantify the relationship between variability in surface slope, wind direction and accumulation rates we use the standard deviation σ of the MSWD and snow accumulation…."

**Specific comments:**

**Line 23: "overall" instead of "entire"?**

*Response:* We have changed the wording.

**Lines 26-28: Implications of this study for developing elevation bias corrections is stated, but I'd encourage the authors to include other broader implications (e.g. for quantification of surface mass balance and for ice core studies) here.**

*Response:* We feel that our section 3.2 already addresses the broader implications. Since we don't present specific results in the paper we feel any statements we could make will be speculative and don't belong into this paper.

**Line 46: "…ice surface, volume…"?**

*Response:* We have inserted a comma after surface.

**Line 47: "Radio-wave signal detection below the noise floor…"? Line 50: "…azimuth dependent elevation ….."?**

*Response:* We have changed line 50. We are not sure what is meant by the comment on line 47.

**Line 52: "…we specifically studied…."?**

*Response:* Using present tense is generally considered better writing style than using past tense and most of our manuscript is written in present tense. We'll leave this up to the copy editor.

**Line 62: Change title to "Data sets and methods"? Line 85: no comma after "both"?**

*Response:* We have deleted the comma in line 85. Section 2 does not contain a description of methods. The methods we apply to the data sets are described in Section 3. We have followed the referee's second broad comment and added a study area section to Section 2.

**Line 86: "…a ground speed…"**

*Response:* we have change "an" to "a".

**Line 105-108: Change to: "The difference in geolocation between distinct elongated topographic snow surface features (sastrugi) between overlapping orthorectified images is on the order of several metres. The DMS images….."?**

*Response:* We have already changed this sentence based on RC1's suggestion.

**Lines 119-123: More detail on the survey flights would be helpful. For example, what was terrain clearance, and was it consistent each year?**

*Response:* We have added a column to Table 1 that shows the flight elevation in meters above ground level (AGL) for each survey flight, which was the same for all 6 flights. Ground speed and other flight parameters are already listed in the text.

**Line 127: Section "3.1 Background": It is unclear to me why this section is titled background, and why it is positioned here. If the section is background, then it should probably come before datasets/methods.**

**This section seems to be a mix of background (lines 128-137 & 150-157) and surface roughness results (137-144).**

*Response:* We have addressed this point in previous comments.

**Line 134: "Elongated….sastrugi" is repetition of line 105-108.**

*Response:* We have already reworded line 134 based on RC1's comments.

**Line 155: Use of Gow reference is really effective. Engagement with more literature of this age could benefit the paper, e.g. publications from the South Pole-Queen Maud Land traverses of 1964-65 and 1965-66. Lots of the results of these traverses are published in "Antarctic Snow and Ice Studies II" https://agupubs.onlinelibrary.wiley.com/doi/book/10.1029/AR016**

*Response:* We have added the SP-QML reference to the new survey area section. The SP-QML traverse has very sparse accumulation measurements but no discussion of roughness or slope to our knowledge. We therefore think it is lacking the relevance to our work that would warrant more discussion. The Gow paper, on the other hand cover all these topics and is therefore relevant.

**Line 158: Section "3.2. Relevance of surface roughness…..": This seems to be background material rather than results.**

*Response:* We have addressed this point in previous comments.

**Line 170: No need for an acronym for bidirectional reflectance distribution function. It is only used once after this in the entire manuscript.**

*Response:* We have deleted the acronym and spelled out BRDF in line 175.

**Line 172: "the relationships"**

*Response:* We prefer "then" since the 2007 work is a follow up to the 2002 work.

**Line 180: Section "3.3. Surface roughness estimates": The entirety of section 3.3 seems to be methods, or assessment of methods (lines 207-216) rather than results or discussion. An overall decision needs to be made about what section 3 is (see 'broad comments' above).**

*Response:* We have addressed this point in previous comments.

**Line 234: year of Mouginot reference (2019?) is missing.**

*Response:* We have added the missing year in the reference.

**Line 239-249: Description of the longitudinal ranges of features of interest. I did find it tricky when reading the text to think about compass bearings in both westerly and easterly compass directions in a single sentence. I understand why the authors have done this (i.e. to describe what is shown in figures 3 and 4), but it is a little jarring and non-intuitive when reading a single sentence. For example, for me it is much easier to comprehend "between 175° W and 60 °E" when written as "between 60-185°E". As currently written, it is also not clear without reference to the figure whether the smooth area is**

**clockwise or anti-clockwise between 175° W and 60 °E. It may also be worth considering annotating the area described in figure 3 (&4)?**

> *Response:* Describing compass bearings in the vicinity of the pole is inherently challenging because of the longitudinal convergence. There are no good solutions to this challenge in our view. The authors are having the same difficulties as the referee when discussing results among ourselves or describing them and we are aware that the readers will face those same challenges. Two conventions are in use to describe longitudes: 180° W/180° E and 0°/360°. Strictly speaking the longitude is defined as an angular measurement ranging from 0° at the Prime Meridian to 180° E eastward and 180° W westward. We have therefore chosen the 180° E/W convention. It also provides a hemisphere distinction that the 0°/360° convention does not have. As the referee points out the words in the text need to be consistent with the figures and we have followed that rule.

> The referee describes the challenges when describing features that go over a singularity. These challenges are the same in both the 180° W/180° E and 0°/360° conventions.

> To help the reader comprehend the spatial setting we have use the same SCAR-recommended polar stereographic map projection (EPSG:3031) for Figs 1-3 that many people are familiar with.

> We are not clear what is meant with the last sentence of the referee's comment.

**Line 250: I am not sure what "…appears to have less of a roughness anomaly…." means. Why not just "..the surface of this dunefield is less rough in 2017 compared to 2014 and 2016"?**

> *Response:* We have changed the sentence accordingly.

**Line 253: "beyond" rather than "out of"?**

> *Response:* We prefer "out of".

**Lines 254-257: this paragraph seems a little 'bolted-on' to this section and is a little perfunctory. Does it need a few more sentences to describe the data presented in figure 4e a little more fully? This section is entitled "3.4 Surface roughness, slope and elevation…" but is very much dominated by roughness.**

> *Response:* We have moved the last sentence into the new survey area section.

**Line 254: Why "…slope of the ATM ICESSN nadir platelets…"? Why not just "ATM-derived surface slope"?**

> *Response:* We prefer to be precise here and state that the slope from the ICESSN data product was used and that we used only the nadir platelet.

**Line 262: A, B & C are labelled in 4b, rather than 4a.**

> *Response:* We have corrected that error.

**Lines 265: "simultaneously" or "concurrently" rather than "at the same time"? Line 267: no comma after "Both"**

*Response:* We have changed line 265 already based on RC1's comments and have deleted the comma after "Both".

**Line 268: "…seems to be even slightly lower…" – recommend rewording to either quantify this statement, or to make it more certain. Perhaps "…are slightly lower…"?**

*Response:* We have reworded the sentence accordingly.

**Lines 282-318: "Section 4" – this all seems to be method description here, rather than description of data or results? Perhaps a re-structuring of the paper is required to make it clear to the reader which sections are methods, results and interpretation? If the manuscript is a methods development paper then that's fine, but that's not the impression currently given by the abstract.**

*Response:* We have addressed this point in previous comments.

**Lines 320-340: "Section 5" – the first part of this section is a mix of background information (i.e. lines 320-329) and further description of methods (i.e. lines 329-340). It does not describe "Spatial variability in snow accumulation rates". I would suggest that the opening line of section 5 is not the best place to state "Accumulation of snow on the Antarctic ice sheet is primarily the result of precipitation of snow". Such a sentence should be on the 1st page of a manuscript. The entirety of Lines 320-324 should be much earlier in the manuscript.**

*Response:* We have moved lines 320 – 326 into the new survey area section. We have revised the title of this section to better reflect its contents.

**Line 347: rather than "several", can the authors provide a range (e.g. 0-3 cm w.e. yr-1)?**

*Response:* We have included a number.

**Line 349: again, here it would be good to quantify the statement made (e.g. "the highest accumulation rates (xx cm w.e. yr-1) near…."**

*Response:* We have included a number.

**Line 352-353: Good to cite original paper locating the bedrock low (i.e. Studinger et al. 2006). Authors could also add an up-to-date reference here to reflect new bed data acquisition in this area. Either Paxman et al., 2019 https://agupubs.onlinelibrary.wiley.com/doi/full/10.1029/2018GC008126 or Morlighem et al., 2019 https://www.nature.com/articles/s41561-019-0510-8 ? Perhaps also move the reference to figure 3a to earlier in the sentence as it only shows surface depression, rather than the subglacial topographic low?**

*Response:* We have added a reference to Morlighem et al. (2020) in the new survey area section. The ice thickness and bedrock data that are relevant to our study are the one that have been collected along our survey line on the 6 flights we discuss. These data sets are shown and referenced in our manuscript.

**Line 354-360: this is an extensive description of previous work that is not directly linked into the data description/interpretation here. Could it be moved to a 'study area' section earlier in the manuscript? It might be more effective there, and can then simply be referred to at its current location?**

*Response:* We have moved some of the wording into the new survey area section.

**Line 362: suggest insert an r2 value after "correlates".**

*Response:* We don't have $r^2$ values calculated for geographic segments and don't think it would make much sense.

**Line 365: "highly variable" – requires some quantification in the text (i.e. range of values should be quoted).**

*Response:* We have quantified the variability.

**Line 369: Reword to "However, several peaks in snow accumulation rate still correlate…"?**

*Response:* We have changed the wording.

**Line 370: again, insert an r2 value after "correlates"? Line 372: "topographic" rather than "topography"?**

*Response:* We have changed the wording. Figure A3 shows a visual correlation without $r^2$ values.

**Line 372-373: change to "..lowest part of the depression where it reaches it's highest point."? Perhaps quantify the "highest point" too? How high was it? Such statements should be quantified in the text.**

*Response:* We have changed the sentence accordingly, but have replaced "it's" with "its".

**Lines 373-374: change to "…with lows in topography results in an overall negative correlation coefficient of…."**

*Response:* We have changed the sentence accordingly.

**Lines 393-394: insert an r2 value after "The correlation is strongest"? I note that in-text quantification of data description is much better in the following section 6.**

*Response:* We don't have $r^2$ values calculated for geographic segments and don't think it would make much sense.

**Line 432: is there really a requirement to say "ESA's CryoSat-2"? Why not just CryoSat-2? Line 466: change to "….MERRA-2, which have low spatial resolutions."?**

*Response:* We have deleted "ESA's" to make this CryoSat-2 mentioning consistent with previous mentioning of CryoSat-2 in the manuscript. We have changed line 466 following the suggestion.

**Lines 467-471: this is a very important finding.**

*Response:* None required

**Figures:**

**Figure 1: Cite source of rock outcrop polygons (Antarctic Digital Database?)**

*Response:* We have added the Antarctic Digital Database to the caption of Fig. 1 and data availability section at the end of the manuscript.

**Figure 2: I found it difficult to orient myself between figures 2 and 3. Where is figure 2 located on figure 3?**

*Response:* Figure 2 is located at 135° E and 88° S. We have added the location of Fig. 2 to Fig. 3b and updated the caption to Fig. 2. The same SCAR-recommended polar stereographic map projection (EPSG:3031) is used for Figs 1-3.

**Table 1: This is table is really useful.**

*Response:* None required.

[revised manuscript text omitted]

---

## Author Response (AR2)

**Response to the Editor's decision (Nanna Bjørnholt Karlsson) on manuscript TC-2020-51: Temporal and spatial variability in surface roughness and accumulation rate around 88° S from repeat airborne geophysical surveys**

We use the following color and font coding scheme in our response:

**Editors's comments**

> *Response:* authors' response to comments.

Dear Dr. Karlsson,

We would like to thank you for your very thorough and thoughtful review of our manuscript and your decision to accept it for publication pending corrections. We have corrected both issues according to your suggestions. Our detailed explanation is below. We have also corrected two minor typos in the manuscript related to references that previously went unnoticed.

Best regards,

Michael Studinger
* * *
**Detailed explanation of corrections:**

**Line 36: Is there a word missing? "The southern convergence of all Ice, Cloud, and land Elevation Satellite-2 (ICESat-2, (Markus et al., 2017)) and CryoSat-2 (Wingham et al., 2006) ground reference tracks at 88°S (McConnell et al., 1997; Mosley-Thompson et al., 1999; Winski et al., 2019, Helm et al., 2014)."**

> *Response:* We have added the missing word: "The southern convergence of all Ice, Cloud, and land Elevation Satellite-2 (ICESat-2, (Markus et al., 2017)) and CryoSat-2 (Wingham et al., 2006) ground reference tracks **is** at 88°S." This sentence was heavily changed in the previous revisions and the references related to accumulation rate and slope are not needed. The accumulation rate and slope statements were moved to lines 72-74 in the previous revisions together with the references.

**From previous editor report: "Fig. 3/line 239: add an arrow to Fig. 3". This is a suggestion to add an arrow indicating where the smooth area located between 175° W and 60° E can be seen on Figure 3. The comment relates back to the difficulty in coupling the information shown in Fig. 3 with Fig. 4. You have already discussed this in your reply to reviewer 1, and I agree with you that regardless of**

**coordinate system it will always be challenging when features cross the 180W/180E line. If you prefer to not annotate your figures further perhaps you can specify somewhere (either in the text or in the figure caption) that Fig. 4 starts at Titan Dome and moves clockwise along 88°S.**

> *Response:* We have marked the smooth segment between 175° W and 60° E with a solid green line in Fig. 3 and have also added a label referring to Fig. 4. We have expanded the figure caption accordingly. We have also indicated the smooth segment with a green line in Fig. 4d, added a label and refer the reader to Fig. 3. We have updated the figure caption accordingly. We have also included your suggestion of pointing out the clockwise orientation in the caption of Fig. 4.
>
> We hope these two additions will make it easier for the reader to locate the smooth segment in Figs. 3 and 4.

[revised manuscript text omitted]